META-RESEARCH ARTICLE

# Assessing the evolution of research topics in a biological field using plant science as an example

**Shin-Han Shiu** [1,2,3]*, **Melissa D. Lehti-Shiu** [1]

**1** Department of Plant Biology, Michigan State University, East Lansing, Michigan, United States of America,
**2** Department of Computational Mathematics, Science, and Engineering, Michigan State University, East Lansing, Michigan, United States of America, **3** DOE-Great Lake Bioenergy Research Center, Michigan State University, East Lansing, Michigan, United States of America

* shius@msu.edu

## Abstract

Scientific advances due to conceptual or technological innovations can be revealed by examining how research topics have evolved. But such topical evolution is difficult to uncover and quantify because of the large body of literature and the need for expert knowledge in a wide range of areas in a field. Using plant biology as an example, we used machine learning and language models to classify plant science citations into topics representing interconnected, evolving subfields. The changes in prevalence of topical records over the last 50 years reflect shifts in major research trends and recent radiation of new topics, as well as turnover of model species and vastly different plant science research trajectories among countries. Our approaches readily summarize the topical diversity and evolution of a scientific field with hundreds of thousands of relevant papers, and they can be applied broadly to other fields.

## Introduction

The explosive growth of scientific data in recent years has been accompanied by a rapidly increasing volume of literature. These records represent a major component of our scientific knowledge and embody the history of conceptual and technological advances in various fields over time. Our ability to wade through these records is important for identifying relevant literature for specific topics, a crucial practice of any scientific pursuit [1]. Classifying the large body of literature into topics can provide a useful means to identify relevant literature. In addition, these topics offer an opportunity to assess how scientific fields have evolved and when major shifts in took place. However, such classification is challenging because the relevant articles in any topic or domain can number in the tens or hundreds of thousands, and the literature is in the form of natural language, which takes substantial effort and expertise to process [2,3]. In addition, even if one could digest all literature in a field, it would still be difficult to quantify such knowledge.

**Data Availability Statement:** The plant science corpus data are available through Zenodo (https://zenodo.org/records/10022686). The codes for the entire project are available through GitHub (https://

github.com/ShiuLab/plant_sci_hist) and Zenodo (https://doi.org/10.5281/zenodo.10894387).

**Funding:** This work was supported by the National Science Foundation (IOS-2107215 and MCB-2210431 to MDL and SHS; DGE-1828149 and IOS-2218206 to SHS), Department of Energy grant Great Lakes Bioenergy Research Center (DE-SC0018409 to SHS). The funders had no role in study design, data collection and analysis, decision to publish, or preparation of the manuscript.

**Competing interests:** The authors have declared that no competing interests exist.

**Abbreviations:** BERT, Bidirectional Encoder Representations from Transformers; br, brassinosteroid; ccTLD, country code Top Level Domain; c-Tf-Idf, class-based Tf-Idf; ChatGPT, Chat Generative Pretrained Transformer; ga, gibberellic acid; LOWESS, locally weighted scatterplot smoothing; MeSH, Medical Subject Heading; SHAP, SHapley Additive exPlanations; SJR, SCImago Journal Rank; Tf-Idf, Term frequency-Inverse document frequency; UMAP, Uniform Manifold Approximation and Projection.

In the last several years, there has been a quantum leap in natural language processing approaches due to the feasibility of building complex deep learning models with highly flexible architectures [4,5]. The development of large language models such as Bidirectional Encoder Representations from Transformers (BERT; [6]) and Chat Generative Pretrained Transformer (ChatGPT; [7]) has enabled the analysis, generation, and modeling of natural language texts in a wide range of applications. The success of these applications is, in large part, due to the feasibility of considering how the same words are used in different contexts when modeling natural language [6]. One such application is topic modeling, the practice of establishing statistical models of semantic structures underlying a document collection. Topic modeling has been proposed for identifying scientific hot topics over time [1], for example, in synthetic biology [8], and it has also been applied to, for example, automatically identify topical scenes in images [9] and social network topics [10], discover gene programs highly correlated with cancer prognosis [11], capture "chromatin topics" that define cell-type differences [12], and investigate relationships between genetic variants and disease risk [13]. Here, we use topic modeling to ask how research topics in a scientific field have evolved and what major changes in the research trends have taken place, using plant science as an example.

## Results

### Plant science corpora allow classification of major research topics

Plant science, broadly defined, is the study of photosynthetic species, their interactions with biotic/abiotic environments, and their applications. For modeling plant science topical evolution, we first identified a collection of plant science documents (i.e., corpus) using a text classification approach. To this end, we first collected over 30 million PubMed records and narrowed down candidate plant science records by searching for those with plant-related terms and taxon names (see **Materials and methods**). Because there remained a substantial number of false positives (i.e., biomedical records mentioning plants in passing), a set of positive plant science examples from the 17 plant science journals with the highest numbers of plant science publications covering a wide range of subfields and a set of negative examples from journals with few candidate plant science records were used to train 4 types of text classification models (see **Materials and methods**). The best text classification model performed well (F1 = 0.96, F1 of a naïve model = 0.5, perfect model = 1) where the positive and negative examples were clearly separated from each other based on prediction probability of the hold-out testing dataset (false negative rate = 2.6%, false positive rate = 5.2%, **S1A and S1B Fig**). The false prediction rate for documents from the 17 plant science journals annotated with the Medical Subject Heading (MeSH) term "Plants" in NCBI was 11.7% (see **Materials and methods**). The prediction probability distribution of positive instances with the MeSH term has an expected left-skew to lower values (**S1C Fig**) compared with the distributions of all positive instances (**S1A Fig**). Thus, this subset with the MeSH term is a skewed representation of articles from these 17 major plant science journals. To further benchmark the validity of the plant science records, we also conducted manual annotation of 100 records where the false positive and false negative rates were 14.6% and 10.6%, respectively (see **Materials and methods**). Using 12 other plant science journals not included as positive examples as benchmarks, the false negative rate was 9.9% (see **Materials and methods**). Considering the range of false prediction rate estimates with different benchmarks, we should emphasize that the model built with the top 17 plant science journals represents a substantial fraction of plant science publications but with biases. Applying the model to the candidate plant science record led to 421,658 positive predictions, hereafter referred to as "plant science records" (**S1D Fig** and **S1 Data**).

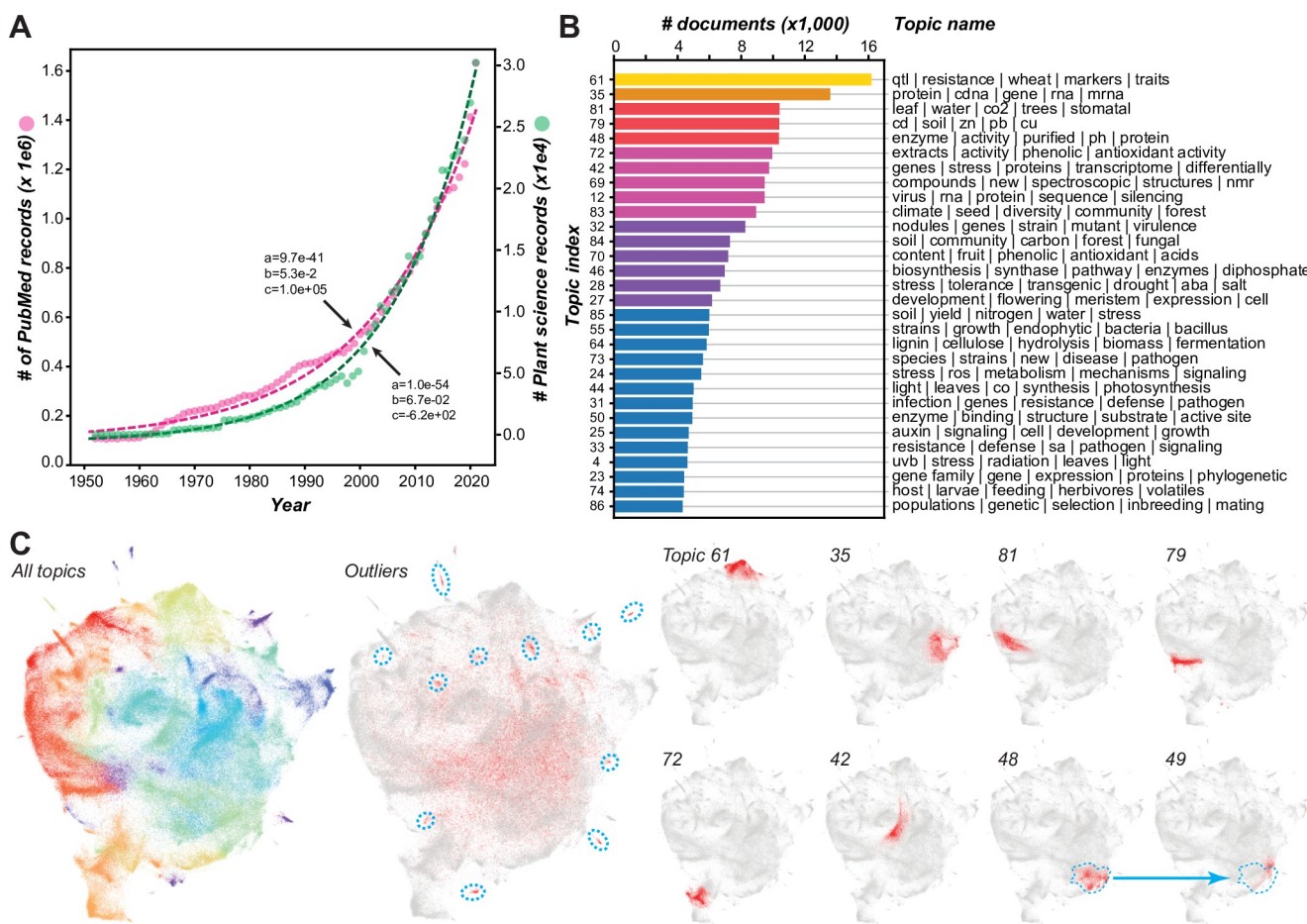

**Fig 1. Change in the number of plant science records over time and diversity of research topics. (A)** Numbers of PubMed (magenta) and plant science (green) records between 1950 and 2020. (a, b, c) Coefficients of the exponential function, $y = ae^b$. Data for the plot are in **S1 Data**. **(B)** Numbers of documents for the top 30 plant science topics. Each topic is designated by an index number (left) and the top 4–6 terms with the highest cTf-Idf values (right). Data for the plot are in **S3 Data**. **(C)** Two-dimensional representation of the relationships between plant science records generated by Uniform Manifold Approximation and Projection (UMAP, [17]) using SciBERT embeddings of plant science records. All topics panel: Different topics are assigned different colors. Outlier panel: UMAP representation of all records (gray) with outlier records in red. Blue dotted circles: areas with relatively high densities indicating topics that are below the threshold for inclusion in a topic. In the 8 UMAP representations on the right, records for example topics are in red and the remaining records in gray. Blue dotted circles indicate the relative position of topic 48.

To better understand how the models classified plant science articles, we identified important terms from a more easily interpretable model (Term frequency-Inverse document frequency (Tf-Idf) model; F1 = 0.934) using Shapley Additive Explanations [14]; 136 terms contributed to predicting plant science records (e.g., Arabidopsis, xylem, seedling) and 138 terms contributed to non-plant science record predictions (e.g., patients, clinical, mice; Tf-Idf feature sheet, **S1 Data**). Plant science records as well as PubMed articles grew exponentially from 1950 to 2020 (**Fig 1A**), highlighting the challenges of digesting the rapidly expanding literature. We used the plant science records to perform topic modeling, which consisted of 4 steps: representing each record as a BERT embedding, reducing dimensionality, clustering, and identifying the top terms by calculating class (i.e., topic)-based Tf-Idf (c-Tf-Idf; [15]). The c-Tf-Idf represents the frequency of a term in the context of how rare the term is to reduce the influence of common words. SciBERT [16] was the best model among those tested (**S2 Data**) and was used for building the final topic model, which classified 372,430 (88.3%) records into 90 topics defined by distinct combinations of terms (**S3 Data**). The topics contained 620 to

16,183 records and were named after the top 4 to 5 terms defining the topical areas (**Fig 1B** and **S3 Data**). For example, the top 5 terms representing the largest topic, topic 61 (16,183 records), are "qtl," "resistance," "wheat," "markers," and "traits," which represent crop improvement studies using quantitative genetics.

Records with assigned topics clustered into distinct areas in a two-dimensional (2D) space (**Fig 1C**, for all topics, see **S4 Data**). The remaining 49,228 outlier records not assigned to any topic (11.7%, middle panel, **Fig 1C**) have 3 potential sources. First, some outliers likely belong to unique topics but have fewer records than the threshold (>500, blue dotted circles, **Fig 1C**). Second, some of the many outliers dispersed within the 2D space (**Fig 1C**) were not assigned to any single topic because they had relatively high prediction scores for multiple topics (**S2 Fig**). These likely represent studies across subdisciplines in plant science. Third, some outliers are likely interdisciplinary studies between plant science and other domains, such as chemistry, mathematics, and physics. Such connections can only be revealed if records from other domains are included in the analyses.

## Topical clusters reveal closely related topics but with distinct key term usage

Related topics tend to be located close together in the 2D representation (e.g., topics 48 and 49, **Fig 1C**). We further assessed intertopical relationships by determining the cosine similarities between topics using cTf-Idfs (**Figs 2A** and **S3**). In this topic network, some topics are closely related and form topic clusters. For example, topics 25, 26, and 27 collectively represent a more general topic related to the field of plant development (cluster *a*, lower left in **Fig 2A**). Other topic clusters represent studies of stress, ion transport, and heavy metals (*b*); photosynthesis, water, and UV-B (*c*); population and community biology (*d*); genomics, genetic mapping, and phylogenetics (*e*, upper right); and enzyme biochemistry (*f*, upper left in **Fig 2A**).

Topics differed in how well they were connected to each other, reflecting how general the research interests or needs are (see **Materials and methods**). For example, topic 24 (stress mechanisms) is the most well connected with median cosine similarity = 0.36, potentially because researchers in many subfields consider aspects of plant stress even though it is not the focus. The least connected topics include topic 21 (clock biology, 0.12), which is surprising because of the importance of clocks in essentially all aspects of plant biology [18]. This may be attributed, in part, to the relatively recent attention in this area.

Examining topical relationships and the cTf-Idf values of terms also revealed how related topics differ. For example, topic 26 is closely related to topics 27 and 25 (cluster *a* on the lower left of **Fig 2A**). Topics 26 and 27 both contain records of developmental process studies mainly in Arabidopsis (**Fig 2B**); however, topic 26 is focused on the impact of light, photoreceptors, and hormones such as gibberellic acids (ga) and brassinosteroids (br), whereas topic 27 is focused on flowering and floral development. Topic 25 is also focused on plant development but differs from topic 27 because it contains records of studies mainly focusing on signaling and auxin with less emphasis on Arabidopsis (**Fig 2C**). These examples also highlight the importance of using multiple top terms to represent the topics. The similarities in cTf-Idfs between topics were also useful for measuring the editorial scope (i.e., diverse, or narrow) of journals publishing plant science papers using a relative topic diversity measure (see **Materials and methods**). For example, *Proceedings of the National Academy of Sciences, USA* has the highest diversity, while *Theoretical and Applied Genetics* has the lowest (**S4 Fig**). One surprise is the relatively low diversity of *American Journal of Botany*, which focuses on plant ecology, systematics, development, and genetics. The low diversity is likely due to the relatively larger

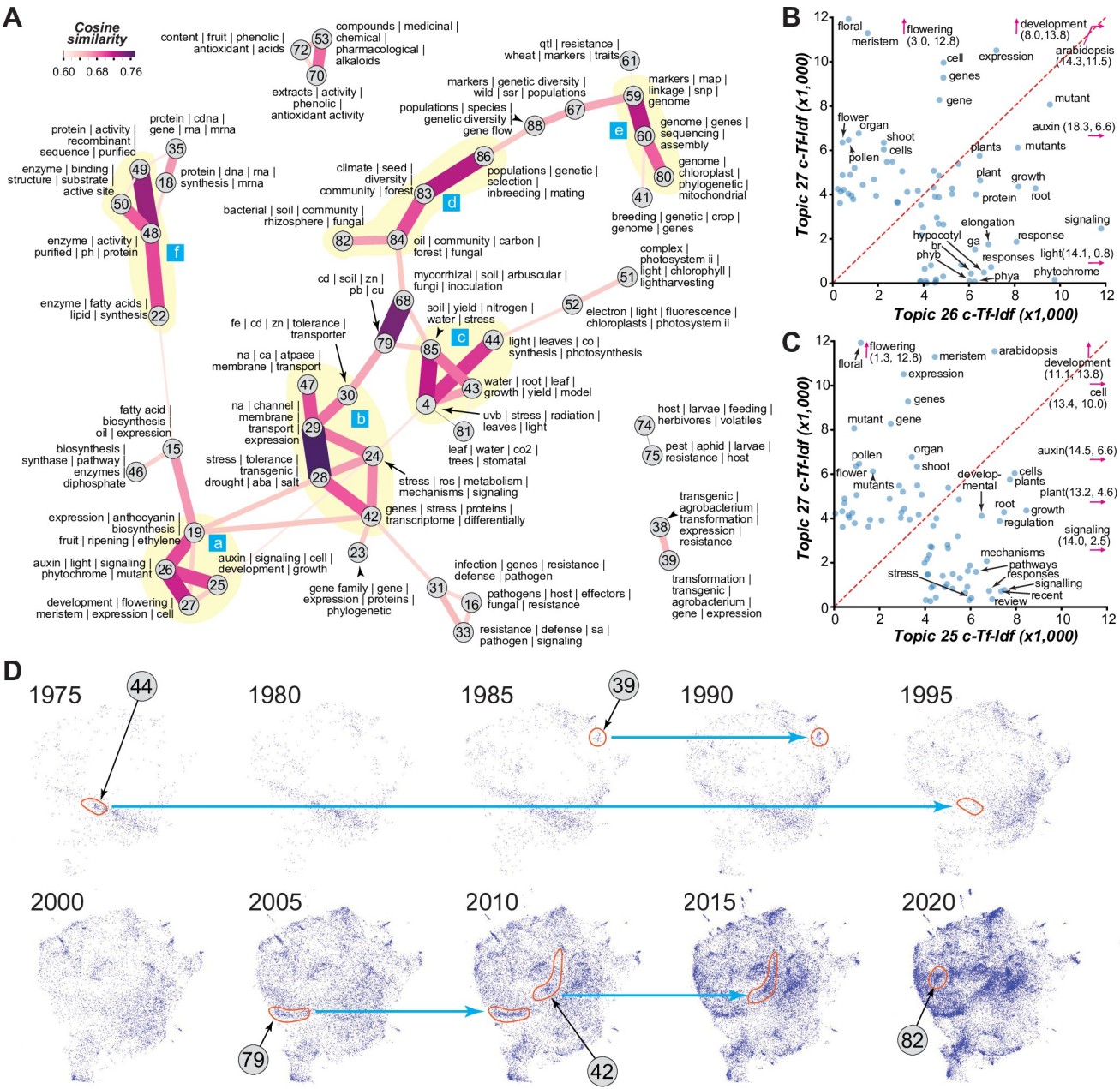

**Fig 2. Intertopical relationships and chronological changes in topical diversity. (A)** Graph depicting the degrees of similarity (edges) between topics (nodes). Between each topic pair, a cosine similarity value was calculated using the cTf-Idf values of all terms. A threshold similarity of 0.6 was applied to illustrate the most related topics. For the full matrix presented as a heatmap, see **S4 Fig**. The nodes are labeled with topic index numbers and the top 4–6 terms. The colors and width of the edges are defined based on cosine similarity. Example topic clusters are highlighted in yellow and labeled a through f (blue boxes). **(B, C)** Relationships between the cTf-Idf values (see **S3 Data**) of the top terms for topics 26 and 27 **(B)** and for topics 25 and 27 **(C)**. Only terms with cTf-Idf ≥ 0.6 are labeled. Terms with cTf-Idf values beyond the x and y axis limit are indicated by pink arrows and cTf-Idf values. **(D)** The 2D representation in **Fig 1C** is partitioned into graphs for different years, and example plots for every 5-year period since 1975 are shown. Example topics discussed in the text are indicated. Blue arrows connect the areas occupied by records of example topics across time periods to indicate changes in document frequencies.

number of cellular and molecular science records in PubMed, consistent with the identification of relatively few topical areas relevant to studies at the organismal, population, community, and ecosystem levels.

## Investigation of the relative prevalence of topics over time reveals topical succession

We next asked whether relationships between topics reflect chronological progression of certain subfields. To address this, we assessed how prevalent topics were over time using dynamic topic modeling [19]. As shown in **Fig 2D**, there is substantial fluctuation in where the records are in the 2D space over time. For example, topic 44 (light, leaves, co, synthesis, photosynthesis) is among the topics that existed in 1975 but has diminished gradually since. In 1985, topic 39 (Agrobacterium-based transformation) became dense enough to be visualized. Additional examples include topics 79 (soil heavy metals), 42 (differential expression), and 82 (bacterial community metagenomics), which became prominent in approximately 2005, 2010, and 2020, respectively (**Fig 2D**). In addition, animating the document occupancy in the 2D space over time revealed a broad change in patterns over time: Some initially dense areas became sparse over time and a large number of topics in areas previously only loosely occupied at the turn of the century increased over time (**S5 Data**).

While the 2D representations reveal substantial details on the evolution of topics, comparison over time is challenging because the number of plant science records has grown exponentially (**Fig 1A**). To address this, the records were divided into 50 chronological bins each with approximately 8,400 records to make cross-bin comparisons feasible (**S6 Data**). We should emphasize that, because of the way the chronological bins were split, the number of records for each topic in each bin should be treated as a normalized value relative to all other topics during the same period. Examining this relative prevalence of topics across bins revealed a clear pattern of topic succession over time (one topic evolved into another) and the presence of 5 topical categories (**Fig 3**). The topics were categorized based on their locally weighted scatterplot smoothing (LOWESS) fits and ordered according to timing of peak frequency (**S7 and S8 Data,** see **Materials and methods**). In **Fig 3**, the relative decrease in document frequency does not mean that research output in a topic is dwindling. Because each row in the heatmap is normalized based on the minimum and maximum values within each topic, there still can be substantial research output in terms of numbers of publications even when the relative frequency is near zero. Thus, a reduced relative frequency of a topic reflects only a below-average growth rate compared with other topical areas.

The first topical category is a stable category with 7 topics mostly established before the 1980s that have since remained stable in terms of prevalence in the plant science records (top of **Fig 3A**). These topics represent long-standing plant science research foci, including studies of plant physiology (topics 4, 58, and 81), genetics (topic 61), and medicinal plants (topic 53). The second category contains 8 topics established before the 1980s that have mostly decreased in prevalence since (the early category, **Fig 3B**). Two examples are physiological and morphological studies of hormone action (topic 45, the second in the early category) and the characterization of protein, DNA, and RNA (topic 18, the second to last). Unlike other early topics, topic 78 (paleobotany and plant evolution studies, the last topic in **Fig 3B**) experienced a resurgence in the early 2000s due to the development of new approaches and databases and changes in research foci [20].

The 33 topics in the third, transitional category became prominent in the 1980s, 1990s, or even 2000s but have clearly decreased in prevalence (**Fig 3C**). In some cases, the early and the transitional topics became less prevalent because of topical succession—refocusing of earlier topics led to newer ones that either show no clear sign of decrease (the sigmoidal category, **Fig 3D**) or continue to increase in prevalence (the rising category, **Fig 3E**). Consistent with the notion of topical succession, topics within each topic cluster (**Fig 2**) were found across topic categories and/or were prominent at different time periods (indicated by colored lines linking

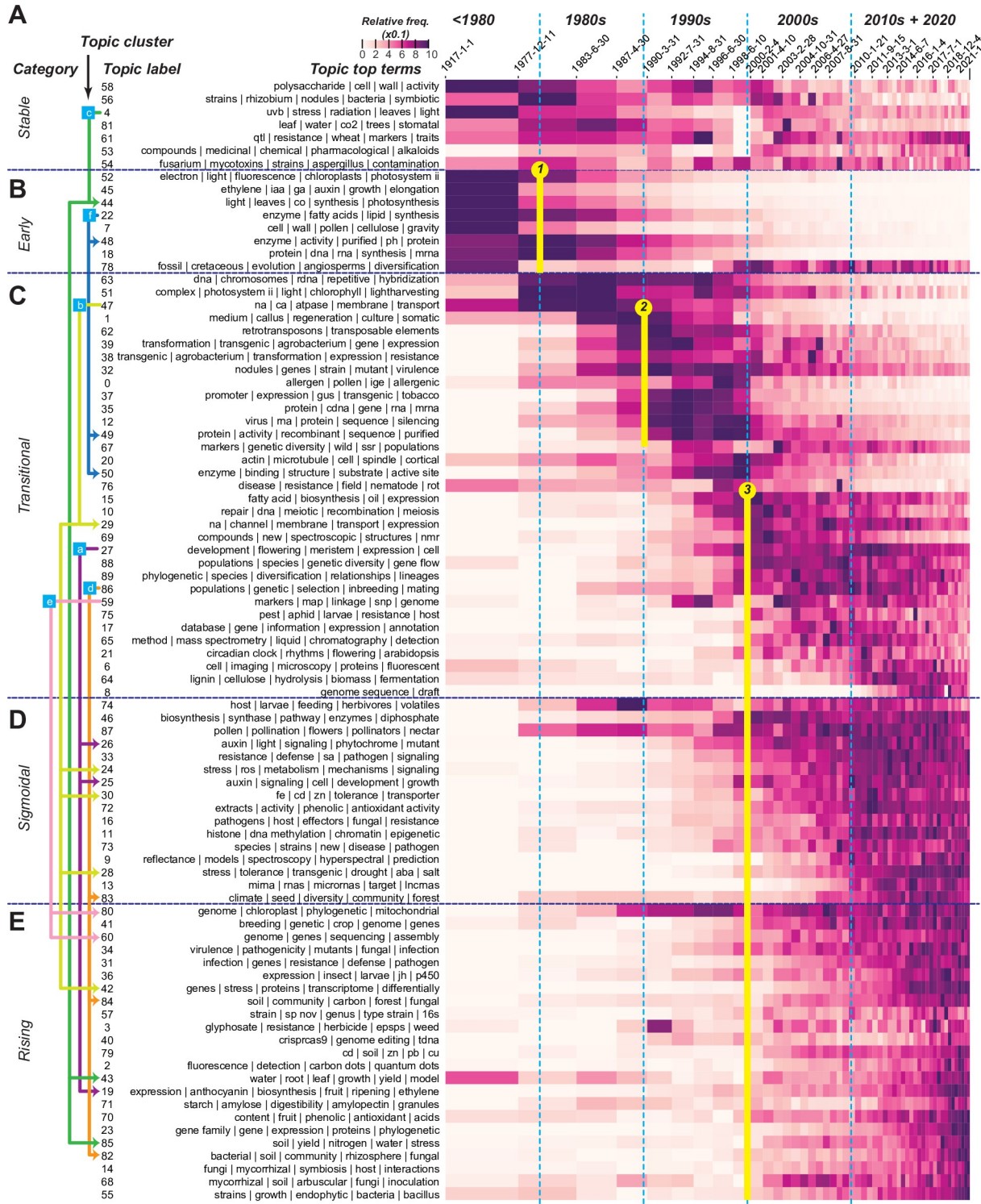

**Fig 3. Topic evolution.** (**A-E**) A heat map of relative topic frequency over time reveals 5 topical categories: (**A**) stable, (**B**) early, (**C**) transitional, (**D**) sigmoidal, and (**E**) rising. The *x* axis denotes different time bins with each bin containing a similar number of documents to account for the exponential growth of plant science records over time. The sizes of all bins except the first are drawn to scale based on the beginning and end dates. The *y* axis lists different topics denoted by the label and top 4 to 5 terms. In each cell, the prevalence of a topic in a time bin is colored according to the min-max normalized cTf-Idf values for that topic. Light blue dotted lines delineate different decades. The arrows left of a subset of topic labels indicate example relationships between topics in topic clusters. Blue boxes with labels *a–f* indicate topic clusters, which are the same as those in **Fig 2**. Connecting lines indicate successional trends. Yellow circles/lines 1–3: 3 major transition patterns. The original data are in **S5 Data**.

topics, **Fig 3**). One example is topics in topic cluster *b* (connected with light green lines and arrows, compare **Figs 2 and 3**); the study of cation transport (topic 47, the third in the transitional category), prominent in the 1980s and early 1990s, is connected to 5 other topics, namely, another transitional topic 29 (cation channels and their expression) peaking in the 2000s and early 2010s, sigmoidal topics 24 and 28 (stress response, tolerance mechanisms) and 30 (heavy metal transport), which rose to prominence in mid-2000s, and the rising topic 42 (stress transcriptomic studies), which increased in prevalence in the mid-2010s.

The rise and fall of topics can be due to a combination of technological or conceptual breakthroughs, maturity of the field, funding constraints, or publicity. The study of transposable elements (topic 62) illustrates the effect of publicity; the rise in this field coincided with Barbara McClintock's 1983 Nobel Prize but not with the publication of her studies in the 1950s [21]. The reduced prevalence in early 2000 likely occurred in part because analysis of transposons became a central component of genome sequencing and annotation studies, rather than dedicated studies. In addition, this example indicates that our approaches, while capable of capturing topical trends, cannot be used to directly infer major papers leading to the growth of a topic.

## Three major topical transition patterns signify shifts in research trends

Beyond the succession of specific topics, 3 major transitions in the dynamic topic graph should be emphasized: (1) the relative decreasing trend of early topics in the late 1970s and early 1980s; (2) the rise of transitional topics in late 1980s; and (3) the relative decreasing trend of transitional topics in the late 1990s and early 2000s, which coincided with a radiation of sigmoidal and rising topics (yellow circles, **Fig 3**). The large numbers of topics involved in these transitions suggest major shifts in plant science research. In transition 1, early topics decreased in relative prevalence in the late 1970s to early 1980s, which coincided with the rise of transitional topics over the following decades (circle 1, **Fig 3**). For example, there was a shift from the study of purified proteins such as enzymes (early topic 48, **S5A Fig**) to molecular genetic dissection of genes, proteins, and RNA (transitional topic 35, **S5B Fig**) enabled by the wider adoption of recombinant DNA and molecular cloning technologies in late 1970s [22]. Transition 2 (circle 2, **Fig 3**) can be explained by the following breakthroughs in the late 1980s: better approaches to create transgenic plants and insertional mutants [23], more efficient creation of mutant plant libraries through chemical mutagenesis (e.g., [24]), and availability of gene reporter systems such as β-glucuronidase [25]. Because of these breakthroughs, molecular genetics studies shifted away from understanding the basic machinery to understanding the molecular underpinnings of specific processes, such as molecular mechanisms of flower and meristem development and the action of hormones such as auxin (topic 27, **S5C Fig**); this type of research was discussed as a future trend in 1988 [26] and remains prevalent to this date. Another example is gene silencing (topic 12), which became a focal area of study along with the widespread use of transgenic plants [27].

Transition 3 is the most drastic: A large number of transitional, sigmoidal, and rising topics became prevalent nearly simultaneously at the turn of the century (circle 3, **Fig 3**). This period also coincides with a rapid increase in plant science citations (**Fig 1A**). The most notable breakthroughs included the availability of the first plant genome in 2000 [28], increasing ease and reduced cost of high-throughput sequencing [29], development of new mass spectrometry–based platforms for analyzing proteins [30], and advancements in microscopic and optical imaging approaches [31]. Advances in genomics and omics technology also led to an increase in stress transcriptomics studies (42, **S5D Fig**) as well as studies in many other topics such as epigenetics (topic 11), noncoding RNA analysis (13), genomics and phylogenetics (80),

breeding (41), genome sequencing and assembly (60), gene family analysis (23), and metagenomics (82 and 55).

In addition to the 3 major transitions across all topics, there were also transitions within topics revealed by examining the top terms for different time bins (heatmaps, **S5 Fig**). Taken together, these observations demonstrate that knowledge about topical evolution can be readily revealed through topic modeling. Such knowledge is typically only available to experts in specific areas and is difficult to summarize manually, as no researcher has a command of the entire plant science literature.

## Analysis of taxa studied reveals changes in research trends

Changes in research trends can also be illustrated by examining changes in the taxa being studied over time (**S9 Data**). There is a strong bias in the taxa studied, with the record dominated by research models and economically important taxa (**S6 Fig**). Flowering plants (Magnoliopsida) are found in 93% of records (**S6A Fig**), and the mustard family Brassicaceae dominates at the family level (**S6B Fig**) because the genus *Arabidopsis* contributes to 13% of plant science records (**Fig 4A**). When examining the prevalence of taxa being studied over time, clear patterns of turnover emerged similar to topical succession (**Figs 4B**, **S6C, and S6D**; **Materials and methods**). Given that Arabidopsis is mentioned in more publications than other species we analyzed, we further examined the trends for Arabidopsis publications. The increase in the normalized number (i.e., relative to the entire plant science corpus) of Arabidopsis records coincided with advocacy of its use as a model system in the late 1980s [32]. While it remains a major plant model, there has been a decrease in overall Arabidopsis publications relative to all other plant science publications since 2011 (blue line, normalized total, **Fig 4C**). Because the same chronological bins, each with same numbers of records, from the topic-over-time analysis (**Fig 3**) were used, the decrease here does not mean that there were fewer Arabidopsis publications—in fact, the number of Arabidopsis papers has remained steady since 2011. This decrease means that Arabidopsis-related publications represent a relatively smaller proportion of plant science records. Interestingly, this decrease took place much earlier (approximately 2005) and was steeper in the United States (red line, **Fig 4C**) than in all countries combined (blue line, **Fig 4C**).

Assuming that the normalized number of publications reflects the relative intensity of research activities, one hypothesis for the relative decrease in focus on Arabidopsis is that advances in, for example, plant transformation, genetic manipulation, and genome research have allowed the adoption of more previously nonmodel taxa. Consistent with this, there was a precipitous increase in the number of genera being published in the mid-90s to early 2000s during which approaches for plant transgenics became established [34], but the number has remained steady since then (**S7A Fig**). The decrease in the proportion of Arabidopsis papers is also negatively correlated with the timing of an increase in the number of draft genomes (**S7B Fig** and **S9 Data**). It is plausible that genome availability for other species may have contributed to a shift away from Arabidopsis. Strikingly, when we analyzed US National Science Foundation records, we found that the numbers of funded grants mentioning Arabidopsis (**S7C Fig**) have risen and fallen in near perfect synchrony with the normalized number of Arabidopsis publication records (red line, **Fig 4C**). This finding likely illustrates the impact of funding on Arabidopsis research.

By considering both taxa information and research topics, we can identify clear differences in the topical areas preferred by researchers using different plant taxa (**Fig 4D** and **S10 Data**). For example, studies of auxin/light signaling, the circadian clock, and flowering tend to be carried out in Arabidopsis, while quantitative genetic studies of disease resistance tend to be done

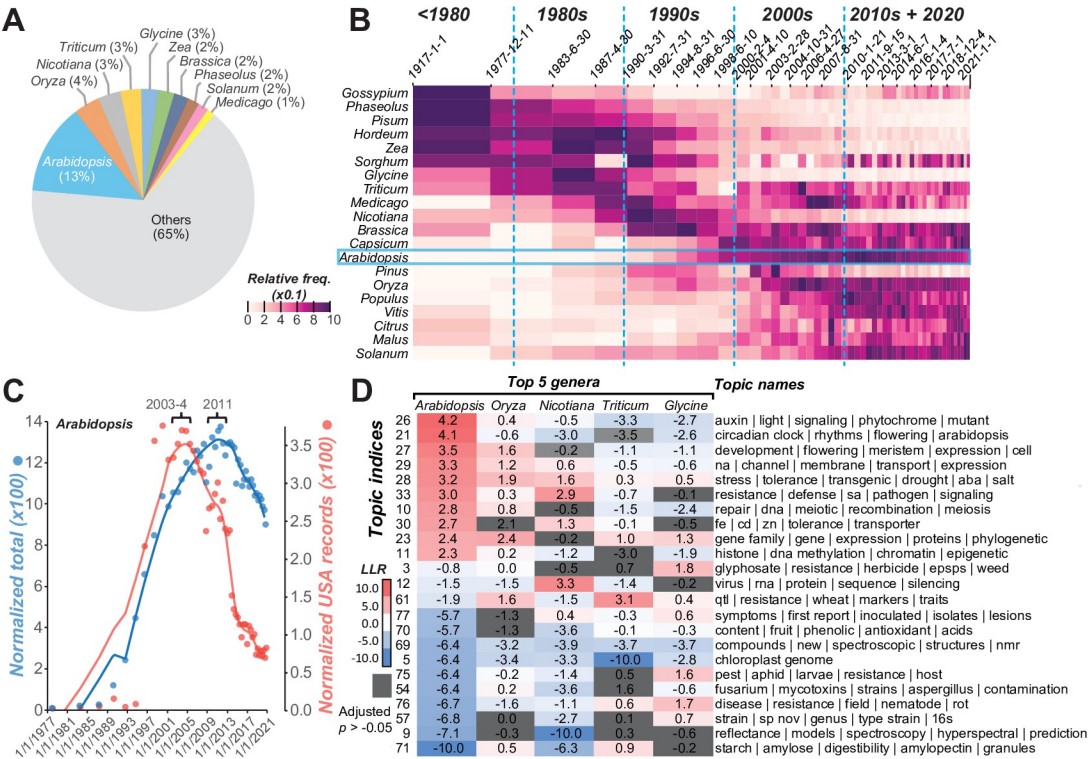

**Fig 4. Prevalence of different plant genera in the plant science records. (A)** Percentage of records mentioning specific genera. **(B)** Change in the prevalence of genera in plant science records over time. **(C)** Changes in the normalized numbers of all records (blue) and records from the US (red) mentioning Arabidopsis over time. The lines are LOWESS fits with fraction parameter = 0.2. **(D)** Topical over (red) and under (blue) representation among 5 genera with the most plant science records. LLR: log 2 likelihood ratios of each topic in each genus. Gray: topic-species combination not significantly enriched at the 5% level based on enrichment *p*-values adjusted for multiple testing with the Benjamini–Hochberg method [33]. The data used for plotting are in **S9 Data**. The statistics for all topics are in **S10 Data**.

in wheat and rice, glyphosate research in soybean, and RNA virus research in tobacco. Taken together, joint analyses of topics and species revealed additional details about changes in preferred models over time, and the preferred topical areas for different taxa.

## Countries differ in their contributions to plant science and topical preference

We next investigated whether there were geographical differences in topical preference among countries by inferring country information from 330,187 records (see **Materials and methods**). The 10 countries with the most records account for 73% of the total, with China and the US contributing to approximately 18% each (**Fig 5A**). The exponential growth in plant science records (green line, **Fig 1A**) was in large part due to the rapid rise in annual record numbers in China and India (**Fig 5B**). When we examined the publication growth rates using the top 17 plant science journals, the general patterns remained the same (**S7D Fig**). On the other hand, the US, Japan, Germany, France, and Great Britain had slower rates of growth compared with all non-top 10 countries. The rapid increase in records from China and India was accompanied by a rapid increase in metrics measuring journal impact (**Figs 5C and S8 and S9 Data**). For example, using citation score (**Fig 5C**, see **Materials and methods**), we found that during a 22-year period China (dark green) and India (light green) rapidly approached the global average (y = 0, yellow), whereas some of the other top 10 countries, particularly the US (red)

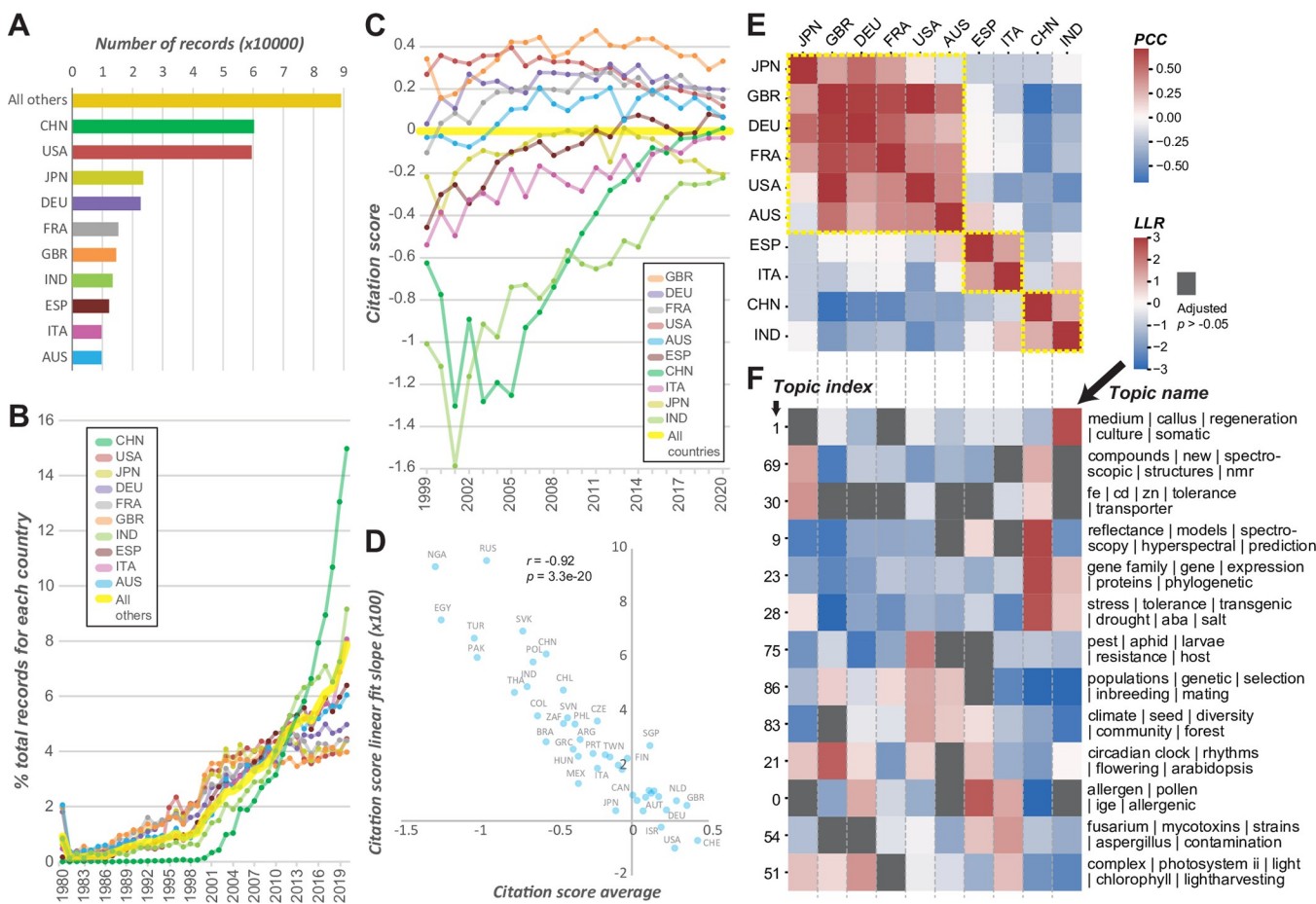

**Fig 5. Plant science record numbers, citation scores, and topical preference of different countries. (A)** Numbers of plant science records for countries with the 10 highest numbers. **(B)** Percentage of all records from each of the top 10 countries from 1980 to 2020. **(C)** Difference in citation scores from 1999 to 2020 for the top 10 countries. **(D)** Shown for each country is the relationship between the citation scores averaged from 1999 to 2020 and the slope of linear fit with year as the predictive variable and citation score as the response variable. The countries with >400 records and with <10% missing impact values are included. Data used for plots **(A–D)** are in **S11 Data**. **(E)** Correlation in topic enrichment scores between the top 10 countries. PCC, Pearson's correlation coefficient, positive in red, negative in blue. Yellow rectangle: countries with more similar topical preferences. **(F)** Enrichment scores (LLR, log likelihood ratio) of selected topics among the top 10 countries. Red: overrepresentation, blue: underrepresentation. Gray: topic-country combination that is not significantly enriched at the 5% level based on enrichment p-values adjusted for multiple testing with the Benjamini–Hochberg method (for all topics and plotting data, see **S12 Data**).

and Japan (yellow green), showed signs of decrease (**Fig 5C**). It remains to be determined whether these geographical trends reflect changes in priority, investment, and/or interest in plant science research.

Interestingly, the relative growth/decline in citation scores over time (measured as the slope of linear fit of year versus citation score) was significantly and negatively correlated with average citation score (**Fig 5D**); i.e., countries with lower overall metrics tended to experience the strongest increase in citation scores over time. Thus, countries that did not originally have a strong influence on plant sciences now have increased impact. These patterns were also observed when using H-index or journal rank as metrics (**S8 Fig** and **S11 Data**) and were not due to increased publication volume, as the metrics were normalized against numbers of records from each country (see **Materials and methods**). In addition, the fact that different metrics with different caveats and assumptions yielded consistent conclusions indicates the robustness of our observations. We hypothesize that this may be a consequence of the ease in scientific communication among geographically isolated research groups. It could also be

because of the prevalence of online journals that are open access, which makes scientific information more readily accessible. Or it can be due to the increasing international collaboration. In any case, the causes for such regression toward the mean are not immediately clear and should be addressed in future studies.

We also assessed how the plant research foci of countries differ by comparing topical preference (i.e., the degree of enrichment of plant science records in different topics) between countries. For example, Italy and Spain cluster together (yellow rectangle, **Fig 5E**) partly because of similar research focusing on allergens (topic 0) and mycotoxins (topic 54) and less emphasis on gene family (topic 23) and stress tolerance (topic 28) studies (**Fig 5F**, for the fold enrichment and corrected *p*-values of all topics, see **S12 Data**). There are substantial differences in topical focus between countries (**S9 Fig**). For example, research on new plant compounds associated with herbal medicine (topic 69) is a focus in China but not in the US, but the opposite is true for population genetics and evolution (topic 86) (**Fig 5F**). In addition to revealing how plant science research has evolved over time, topic modeling provides additional insights into differences in research foci among different countries, which are informative for science policy considerations.

## Discussion

In this study, topic modeling revealed clear transitions among research topics, which represent shifts in research trends in plant sciences. One limitation of our study is the bias in the PubMed-based corpus. The cellular, molecular, and physiological aspects of plant sciences are well represented, but there are many fewer records related to evolution, ecology, and systematics. Our use of titles/abstracts from the top 17 plant science journals as positive examples allowed us to identify papers we typically see in these journals, but this may have led to us missing "outlier" articles, which may be the most exciting. Another limitation is the need to assign only one topic to a record when a study is interdisciplinary and straddles multiple topics. Furthermore, a limited number of large, inherently heterogeneous topics were summarized to provide a more concise interpretation, which undoubtedly underrepresents the diversity of plant science research. Despite these limitations, dynamic topic modeling revealed changes in plant science research trends that coincide with major shifts in biological science. While we were interested in identifying conceptual advances, our approach can identify the trend but the underlying causes for such trends, particularly key records leading to the growth in certain topics, still need to be identified. It also remains to be determined which changes in research trends lead to paradigm shifts as defined by Kuhn [35].

The key terms defining the topics frequently describe various technologies (e.g., topic 38/39: transformation, 40: genome editing, 59: genetic markers, 65: mass spectrometry, 69: nuclear magnetic resonance) or are indicative of studies enabled through molecular genetics and omics technologies (e.g., topic 8/60: genome, 11: epigenetic modifications, 18: molecular biological studies of macromolecules, 13: small RNAs, 61: quantitative genetics, 82/84: metagenomics). Thus, this analysis highlights how technological innovation, particularly in the realm of omics, has contributed to a substantial number of research topics in the plant sciences, a finding that likely holds for other scientific disciplines. We also found that the pattern of topic evolution is similar to that of succession, where older topics have mostly decreased in relative prevalence but appear to have been superseded by newer ones. One example is the rise of transcriptome-related topics and the correlated, reduced focus on regulation at levels other than transcription. This raises the question of whether research driven by technology negatively impacts other areas of research where high-throughput studies remain challenging.

One observation on the overall trends in plant science research is the approximately 10-year cycle in major shifts. One hypothesis is related to not only scientific advances but also

to the fashion-driven aspect of science. Nonetheless, given that there were only 3 major shifts and the sample size is small, it is difficult to speculate as to why they happened. By analyzing the country of origin, we found that China and India have been the 2 major contributors to the growth in the plant science records in the last 20 years. Our findings also show an equalizing trend in global plant science where countries without a strong plant science publication presence have had an increased impact over the last 20 years. In addition, we identified significant differences in research topics between countries reflecting potential differences in investment and priorities. Such information is important for discerning differences in research trends across countries and can be considered when making policy decisions about research directions.

## Materials and methods

### Collection and preprocessing of a candidate plant science corpus

For reproducibility purposes, a random state value of 20220609 was used throughout the study. The PubMed baseline files containing citation information (ftp://ftp.ncbi.nlm.nih.gov/pubmed/baseline/) were downloaded on November 11, 2021. To narrow down the records to plant science-related citations, a candidate citation was identified as having, within the titles and/or abstracts, at least one of the following words: "plant," "plants," "botany," "botanical," "planta," and "plantarum" (and their corresponding upper case and plural forms), or plant taxon identifiers from NCBI Taxonomy (https://www.ncbi.nlm.nih.gov/taxonomy) or USDA PLANTS Database (https://plants.sc.egov.usda.gov/home). Note the search terms used here have nothing to do with the values of the keyword field in PubMed records. The taxon identifiers include all taxon names including and at taxonomic levels below "Viridiplantae" till the genus level (species names not used). This led to 51,395 search terms. After looking for the search terms, qualified entries were removed if they were duplicated, lacked titles and/or abstracts, or were corrections, errata, or withdrawn articles. This left 1,385,417 citations, which were considered the candidate plant science corpus (i.e., a collection of texts). For further analysis, the title and abstract for each citation were combined into a single entry. Text was preprocessed by lowercasing, removing stop-words (i.e., common words), removing non-alphanumeric and non-white space characters (except Greek letters, dashes, and commas), and applying lemmatization (i.e., grouping inflected forms of a word as a single word) for comparison. Because lemmatization led to truncated scientific terms, it was not included in the final preprocessing pipeline.

### Definition of positive/negative examples

Upon closer examination, a large number of false positives were identified in the candidate plant science records. To further narrow down citations with a plant science focus, text classification was used to distinguish plant science and non-plant science articles (see next section). For the classification task, a negative set (i.e., non-plant science citations) was defined as entries from 7,360 journals that appeared <20 times in the filtered data (total = 43,329, journal candidate count, **S1 Data**). For the positive examples (i.e., true plant science citations), 43,329 plant science citations (positive examples) were sampled from 17 established plant science journals each with >2,000 entries in the filtered dataset: "Plant physiology," "Frontiers in plant science," "Planta," "The Plant journal: for cell and molecular biology," "Journal of experimental botany," "Plant molecular biology," "The New phytologist," "The Plant cell," "Phytochemistry," "Plant & cell physiology," "American journal of botany," "Annals of botany," "BMC plant biology," "Tree physiology," "Molecular plant-microbe interactions: MPMI," "Plant biology," and "Plant biotechnology journal" (journal candidate count, **S1 Data**). Plant biotechnology

journal was included, but only 1,894 records remained after removal of duplicates, articles with missing info, and/or withdrawn articles. The positive and negative sets were randomly split into training and testing subsets (4:1) while maintaining a 1:1 positive-to-negative ratio.

## Text classification based on Tf and Tf-Idf

Instead of using the preprocessed text as features for building classification models directly, text embeddings (i.e., representations of texts in vectors) were used as features. These embeddings were generated using 4 approaches (model summary, **S1 Data**): Term-frequency (Tf), Tf-Idf [36], Word2Vec [37], and BERT [6]. The Tf- and Tf-Idf-based features were generated with CountVectorizer and TfidfVectorizer, respectively, from Scikit-Learn [38]. Different maximum features (1e4 to 1e5) and n-gram ranges (uni-, bi-, and tri-grams) were tested. The features were selected based on the $p$-value of chi-squared tests testing whether a feature had a higher-than-expected value among the positive or negative classes. Four different $p$-value thresholds were tested for feature selection. The selected features were then used to retrain vectorizers with the preprocessed training texts to generate feature values for classification. The classification model used was XGBoost [39] with 5 combinations of the following hyperparameters tested during 5-fold stratified cross-validation: min_child_weight = (1, 5, 10), gamma = (0.5, 1, 1.5, 2.5), subsample = (0.6, 0.8, 1.0), colsample_bytree = (0.6, 0.8, 1.0), and max_depth = (3, 4, 5). The rest of the hyperparameters were held constant: learning_rate = 0.2, n_estimators = 600, objective = binary:logistic. RandomizedSearchCV from Scikit-Learn was used for hyperparameter tuning and cross-validation with scoring = F1-score.

Because the Tf-Idf model had a relatively high model performance and was relatively easy to interpret (terms are frequency-based, instead of embedding-based like those generated by Word2Vec and BERT), the Tf-Idf model was selected as input to SHapley Additive exPlanations (SHAP; [14]) to assess the importance of terms. Because the Tf-Idf model was based on XGBoost, a tree-based algorithm, the TreeExplainer module in SHAP was used to determine a SHAP value for each entry in the training dataset for each Tf-Idf feature. The SHAP value indicates the degree to which a feature positively or negatively affects the underlying prediction. The importance of a Tf-Idf feature was calculated as the average SHAP value of that feature among all instances. Because a Tf-Idf feature is generated based on a specific term, the importance of the Tf-Idf feature indicates the importance of the associated term.

## Text classification based on Word2Vec

The preprocessed texts were first split into train, validation, and test subsets (8:1:1). The texts in each subset were converted to 3 n-gram lists: a unigram list obtained by splitting tokens based on the space character, or bi- and tri-gram lists built with Gensim [40]. Each n-gram list of the training subset was next used to fit a Skip-gram Word2Vec model with vector_size = 300, window = 8, min_count = (5, 10, or 20), sg = 1, and epochs = 30. The Word2Vec model was used to generate word embeddings for train, validate, and test subsets. In the meantime, a tokenizer was trained with train subset unigrams using Tensorflow [41] and used to tokenize texts in each subset and turn each token into indices to use as features for training text classification models. To ensure all citations had the same number of features (500), longer texts were truncated, and shorter ones were zero-padded. A deep learning model was used to train a text classifier with an input layer the same size as the feature number, an attention layer incorporating embedding information for each feature, 2 bidirectional Long-Short-Term-Memory layers (15 units each), a dense layer (64 units), and a final, output layer with 2 units. During training, adam, accuracy, and sparse_categorical_crossentropy were used as the optimizer, evaluation metric, and loss function, respectively. The training process lasted 30 epochs with early

stopping if validation loss did not improve in 5 epochs. An F1 score was calculated for each n-gram list and min_count parameter combination to select the best model (model summary, **S1 Data**).

## Text classification based on BERT models

Two pretrained models were used for BERT-based classification: DistilBERT (Hugging face repository [42] model name and version: distilbert-base-uncased [43]) and SciBERT (allenai/scibert-scivocab-uncased [16]). In both cases, tokenizers were retrained with the training data. BERT-based models had the following architecture: the token indices (512 values for each token) and associated masked values as input layers, pretrained BERT layer (512 × 768) excluding outputs, a 1D pooling layer (768 units), a dense layer (64 units), and an output layer (2 units). The rest of the training parameters were the same as those for Word2Vec-based models, except training lasted for 20 epochs. Cross-validation F1-scores for all models were compared and used to select the best model for each feature extraction method, hyperparameter combination, and modeling algorithm or architecture (model summary, **S1 Data**). The best model was the Word2Vec-based model (min_count = 20, window = 8, ngram = 3), which was applied to the candidate plant science corpus to identify a set of plant science citations for further analysis. The candidate plant science records predicted as being in the positive class (421,658) by the model were collectively referred to as the "plant science corpus."

## Plant science record classification

In PubMed, 1,384,718 citations containing "plant" or any plant taxon names (from the phylum to genus level) were considered candidate plant science citations. To further distinguish plant science citations from those in other fields, text classification models were trained using titles and abstracts of positive examples consisting of citations from 17 plant science journals, each with >2,000 entries in PubMed, and negative examples consisting of records from journals with fewer than 20 entries in the candidate set. Among 4 models tested the best model (built with Word2Vec embeddings) had a cross validation F1 of 0.964 (random guess F1 = 0.5, perfect model F1 = 1, **S1 Data**). When testing the model using 17,330 testing set citations independent from the training set, the F1 remained high at 0.961.

We also conducted another analysis attempting to use the MeSH term "Plants" as a benchmark. Records with the MeSH term "Plants" also include pharmaceutical studies of plants and plant metabolites or immunological studies of plants as allergens in journals that are not generally considered plant science journals (e.g., *Acta astronautica*, *International journal for parasitology*, *Journal of chromatography*) or journals from local scientific societies (e.g., *Acta pharmaceutica Hungarica*, *Huan jing ke xue*, *Izvestiia Akademii nauk. Seriia biologicheskaia*). Because we explicitly labeled papers from such journals as negative examples, we focused on 4,004 records with the "Plants" MeSH term published in the 17 plant science journals that were used as positive instances and found that 88.3% were predicted as the positive class. Thus, based on the MeSH term, there is an 11.7% false prediction rate.

We also enlisted 5 plant science colleagues (3 advanced graduate students in plant biology and genetic/genome science graduate programs, 1 postdoctoral breeder/quantitative biologist, and 1 postdoctoral biochemist/geneticist) to annotate 100 randomly selected abstracts as a reviewer suggested. Each record was annotated by 2 colleagues. Among 85 entries where the annotations are consistent between annotators, 48 were annotated as negative but with 7 predicted as positive (false positive rate = 14.6%) and 37 were annotated as positive but with 4 predicted as negative (false negative rate = 10.8%). To further benchmark the performance of the text classification model, we identified another 12 journals that focus on plant science studies

to use as benchmarks: *Current opinion in plant biology* (number of articles: 1,806), *Trends in plant science* (1,723), *Functional plant biology* (1,717), *Molecular plant pathology* (1,573), *Molecular plant* (1,141), *Journal of integrative plant biology* (1,092), *Journal of plant research* (1,032), *Physiology and molecular biology of plants* (830), *Nature plants* (538), *The plant pathology journal* (443). *Annual review of plant biology* (417), and *The plant genome* (321). Among the 12,611 candidate plant science records, 11,386 were predicted as positive. Thus, there is a 9.9% false negative rate.

## Global topic modeling

BERTopic [15] was used for preliminary topic modeling with n-grams = (1,2) and with an embedding initially generated by DistilBERT, SciBERT, or BioBERT (dmis-lab/biobert-base-cased-v1.2; [44]). The embedding models converted preprocessed texts to embeddings. The topics generated based on the 3 embeddings were similar (**S2 Data**). However, SciBERT-, Bio-BERT-, and distilBERT-based embedding models had different numbers of outlier records (268,848, 293,790, and 323,876, respectively) with topic index = −1. In addition to generating the fewest outliers, the SciBERT-based model led to the highest number of topics. Therefore, SciBERT was chosen as the embedding model for the final round of topic modeling. Modeling consisted of 3 steps. First, document embeddings were generated with SentenceTransformer [45]. Second, a clustering model to aggregate documents into clusters using hdbscan [46] was initialized with min_cluster_size = 500, metric = euclidean, cluster_selection_method = eom, min_samples = 5. Third, the embedding and the initialized hdbscan model were used in BERTopic to model topics with neighbors = 10, nr_topics = 500, ngram_range = (1,2). Using these parameters, 90 topics were identified. The initial topic assignments were conservative, and 241,567 records were considered outliers (i.e., documents not assigned to any of the 90 topics). After assessing the prediction scores of all records generated from the fitted topic models, the 95-percentile score was 0.0155. This score was used as the threshold for assigning outliers to topics: If the maximum prediction score was above the threshold and this maximum score was for topic $t$, then the outlier was assigned to $t$. After the reassignment, 49,228 records remained outliers. To assess if some of the outliers were not assigned because they could be assigned to multiple topics, the prediction scores of the records were used to put records into 100 clusters using $k$-means. Each cluster was then assessed to determine if the outlier records in a cluster tended to have higher prediction scores across multiple topics (**S2 Fig**).

## Topics that are most and least well connected to other topics

The most well-connected topics in the network include topic 24 (stress mechanisms, median cosine similarity = 0.36), topic 42 (genes, stress, and transcriptomes, 0.34), and topic 35 (molecular genetics, 0.32, all $t$ test $p$-values $< 1 \times 10^{-22}$). The least connected topics include topic 0 (allergen research, median cosine similarity = 0.12), topic 21 (clock biology, 0.12), topic 1 (tissue culture, 0.15), and topic 69 (identification of compounds with spectroscopic methods, 0.15; all $t$ test $p$-values $< 1 \times 10^{-24}$). Topics 0, 1, and 69 are specialized topics; it is surprising that topic 21 is not as well connected as explained in the main text.

## Analysis of documents based on the topic model

From the fitted topic model, we obtained a topic-by-term matrix with 91 rows, where each row corresponded to a topic or outlier, and $1.85 \times 10^7$ columns, where each column corresponded to a term in the plant science record vocabulary. This topic-by-term matrix was filled with c-Tf-Idf values for each topic-term combination. The similarity between each pair of topics, excluding the outlier rows, was determined by determining the cosine similarity using the

c-Tf-Idf vectors of the 2 topics in question (**S4 Fig**). Topics with cosine similarities ≥0.6 were connected in the network graph shown in **Fig 2**. The relative topical diversity of a journal $j$ ($v_j$) was calculated as follows:

$$v_j = 1/(var(c_j n_{j,t}/n_{j,a})s)$$

where *var* is variance, $c_j$ is the median cosine similarity between a topic $t$ and non-$t$ topics, $n_{j,t}$ is the number of records of journal $j$ in topic $t$, $n_{j,a}$ is the number of all records of journal $j$ in the plant science corpus, and $s$ is a scaling factor, which is set to $1 \times 10^4$ so that the relative topical diversity value is between 0 and 10 for a more straightforward comparison. The top 50 terms for each topic are available in **S3 Data**. To examine the relationships between documents in a scatter plot with 2 dimensions, we needed to reduce the dimensionality of BERT embeddings (containing information on how 2 documents were related to each other) from 768 to 2. To accomplish this, Uniform Manifold Approximation and Projection (UMAP; [17]) was used. The UMAP parameters evaluated were n_neighbors = (5, 10, 15, 40), min_dist = (0, 0.1, 0.25), and metric = (cosine, euclidean, canberra, mahalanobis, correlation). After comparing the graphs generated with different UMAP parameter combinations, we used the parameter combination n_neighbors = 24, min_dist = 0.1, and metric = cosine to generate the final 2D representation (**S4 Data**).

## Topical diversity among top journals with the most plant science records

Using a relative topic diversity measure (ranging from 0 to 10), we found that there was a wide range of topical diversity among 20 journals with the largest numbers of plant science records (**S3 Fig**). The 4 journals with the highest relative topical diversities are *Proceedings of the National Academy of Sciences*, *USA* (9.6), *Scientific Reports* (7.1), *Plant Physiology* (6.7), and *PLOS ONE* (6.4). The high diversities are consistent with the broad, editorial scopes of these journals. The 4 journals with the lowest diversities are *American Journal of Botany* (1.6), *Oecologia* (0.7), *Plant Disease* (0.7), and *Theoretical and Applied Genetics* (0.3), which reflects their discipline-specific focus and audience of classical botanists, ecologists, plant pathologists, and specific groups of geneticists.

## Dynamic topic modeling

The codes for dynamic modeling were based on _topic_over_time.py in BERTopics and modified to allow additional outputs for debugging and graphing purposes. The plant science citations were binned into 50 subsets chronologically (for timestamps of bins, see **S5 Data**). Because the numbers of documents increased exponentially over time, instead of dividing them based on equal-sized time intervals, which would result in fewer records at earlier time points and introduce bias, we divided them into time bins of similar size (approximately 8,400 documents). Thus, the earlier time subsets had larger time spans compared with later time subsets. If equal-size time intervals were used, the numbers of documents between the intervals would differ greatly; the earlier time points would have many fewer records, which may introduce bias. Prior to binning the subsets, the publication dates were converted to UNIX time (timestamp) in seconds; the plant science records start in 1917-11-1 (timestamp = −1646247600.0) and end in 2021-1-1 (timestamp = 1609477201). The starting dates and corresponding timestamps for the 50 subsets including the end date are in **S6 Data**. The input data included the preprocessed texts, topic assignments of records from global topic modeling, and the binned timestamps of records. Three additional parameters were set for topics_over_time, namely, nr_bin = 50 (number of bins), evolution_tuning = True, and global_tuning = False. The evolution_tuning parameter specified that averaged c-Tf-Idf values for a topic be

calculated in neighboring time bins to reduce fluctuation in c-Tf-Idf values. The global_tuning parameter was set to False because of the possibility that some nonexisting terms could have a high c-Tf-Idf for a time bin simply because there was a high global c-Tf-Idf value for that term.

The binning strategy based on similar document numbers per bin allowed us to increase signal particularly for publications prior to the 90s. This strategy, however, may introduce more noise for bins with smaller time durations (i.e., more recent bins) because of publication frequencies (there can be seasonal differences in the number of papers published, biased toward, e.g., the beginning of the year or the beginning of a quarter). To address this, we examined the relative frequencies of each topic over time (**S7 Data**), but we found that recent time bins had similar variances in relative frequencies as other time bins. We also moderated the impact of variation using LOWESS (10% to 30% of the data points were used for fitting the trend lines) to determine topical trends for **Fig 3**. Thus, the influence of the noise introduced via our binning strategy is expected to be minimal.

## Topic categories and ordering

The topics were classified into 5 categories with contrasting trends: stable, early, transitional, sigmoidal, and rising. To define which category a topic belongs to, the frequency of documents over time bins for each topic was analyzed using 3 regression methods. We first tried 2 forecasting methods: recursive autoregressor (the ForecasterAutoreg class in the skforecast package) and autoregressive integrated moving average (ARIMA implemented in the pmdarima package). In both cases, the forecasting results did not clearly follow the expected trend lines, likely due to the low numbers of data points (relative frequency values), which resulted in the need to extensively impute missing data. Thus, as a third approach, we sought to fit the trendlines with the data points using LOWESS (implemented in the statsmodels package) and applied additional criteria for assigning topics to categories. When fitting with LOWESS, 3 fraction parameters (frac, the fraction of the data used when estimating each y-value) were evaluated (0.1, 0.2, 0.3). While frac = 0.3 had the smallest errors for most topics, in situations where there were outliers, frac = 0.2 or 0.1 was chosen to minimize mean squared errors (**S7 Data**).

The topics were classified into 5 categories based on the slopes of the fitted line over time: (1) stable: topics with near 0 slopes over time; (2) early: topics with negative ($<-0.5$) slopes throughout (with the exception of topic 78, which declined early on but bounced back by the late 1990s); (3) transitional: early positive ($>0.5$) slopes followed by negative slopes at later time points; (4) sigmoidal: early positive slopes followed by zero slopes at later time points; and (5) rising: continuously positive slopes. For each topic, the LOWESS fits were also used to determine when the relative document frequency reached its peak, first reaching a threshold of 0.6 (chosen after trial and error for a range of 0.3 to 0.9), and the overall trend. The topics were then ordered based on (1) whether they belonged to the stable category or not; (2) whether the trends were decreasing, stable, or increasing; (3) the time the relative document frequency first reached 0.6; and (4) the time that the overall peak was reached (**S8 Data**).

## Taxa information

To identify a taxon or taxa in all plant science records, NCBI Taxonomy taxdump datasets were downloaded from the NCBI FTP site (https://ftp.ncbi.nlm.nih.gov/pub/taxonomy/new_taxdump/) on September 20, 2022. The highest-level taxon was Viridiplantae, and all its child taxa were parsed and used as queries in searches against the plant science corpus. In addition, a species-over-time analysis was conducted using the same time bins as used for dynamic topic models. The number of records in different time bins for top taxa are in the genus, family,

order, and additional species level sheet in **S9 Data**. The degree of over-/underrepresentation of a taxon X in a research topic T was assessed using the *p*-value of a Fisher's exact test for a 2 × 2 table consisting of the numbers of records in both X and T, in X but not T, in T but not X, and in neither (**S10 Data**).

For analysis of plant taxa with genome information, genome data of taxa in Viridiplantae were obtained from the NCBI Genome data-hub (https://www.ncbi.nlm.nih.gov/data-hub/genome) on October 28, 2022. There were 2,384 plant genome assemblies belonging to 1,231 species in 559 genera (genome assembly sheet, **S9 Data**). The date of the assembly was used as a proxy for the time when a genome was sequenced. However, some species have updated assemblies and have more recent data than when the genome first became available.

## Taxa being studied in the plant science records

Flowering plants (Magnoliopsida) are found in 93% of records, while most other lineages are discussed in <1% of records, with conifers and related species being exceptions (Acrogynomsopermae, 3.5%, **S6A Fig**). At the family level, the mustard (Brassicaceae), grass (Poaceae), pea (Fabaceae), and nightshade (Solanaceae) families are in 51% of records (**S6B Fig**). The prominence of the mustard family in plant science research is due to the *Brassica* and *Arabidopsis* genera (**Fig 4A**). When examining the prevalence of taxa being studied over time, clear patterns of turnovers emerged (**Figs 4B, S6C, and S6D**). While the study of monocot species (Liliopsida) has remained steady, there was a significant uptick in the prevalence of eudicot (eudicotyledon) records in the late 90s (**S6C Fig**), which can be attributed to the increased number of studies in the mustard, myrtle (Myrtaceae), and mint (Lamiaceae) families among others (**S6D Fig**). At the genus level, records mentioning *Gossypium* (cotton), *Phaseolus* (bean), *Hordeum* (wheat), and *Zea* (corn), similar to the topics in the early category, were prevalent till the 1980s or 1990s but have mostly decreased in number since (**Fig 4B**). In contrast, *Capsicum*, *Arabidopsis*, *Oryza*, *Vitus*, and *Solanum* research has become more prevalent over the last 20 years.

## Geographical information for the plant science corpus

The geographical information (country) of authors in the plant science corpus was obtained from the address (AD) fields of first authors in Medline XML records accessible through the NCBI EUtility API (https://www.ncbi.nlm.nih.gov/books/NBK25501/). Because only first author affiliations are available for records published before December 2014, only the first author's location was considered to ensure consistency between records before and after that date. Among the 421,658 records in the plant science corpus, 421,585 had Medline records and 421,276 had unique PMIDs. Among the records with unique PMIDs, 401,807 contained address fields. For each of the remaining records, the AD field content was split into tokens with a "," delimiter, and the token likely containing geographical info (referred to as location tokens) was selected as either the last token or the second to last token if the last token contained "@" indicating the presence of an email address. Because of the inconsistency in how geographical information was described in the location tokens (e.g., country, state, city, zip code, name of institution, and different combinations of the above), the following 4 approaches were used to convert location tokens into countries.

The first approach was a brute force search where full names and alpha-3 codes of current countries (ISO 3166–1), current country subregions (ISO 3166–2), and historical country (i.e., country that no longer exists, ISO 3166–3) were used to search the address fields. To reduce false positives using alpha-3 codes, a space prior to each code was required for the match. The first approach allowed the identification of 361,242, 16,573, and 279,839 records with current

country, historical country, and subregion information, respectively. The second method was the use of a heuristic based on common address field structures to identify "location strings" toward the end of address fields that likely represent countries, then the use of the Python pycountry module to confirm the presence of country information. This approach led to 329,025 records with country information. The third approach was to parse first author email addresses (90,799 records), recover top-level domain information, and use country code Top Level Domain (ccTLD) data from the ISO 3166 Wikipedia page to define countries (72,640 records). Only a subset of email addresses contains country information because some are from companies (.com), nonprofit organizations (.org), and others. Because a large number of records with address fields still did not have country information after taking the above 3 approaches, another approach was implemented to query address fields against a locally installed Nominatim server (v.4.2.3, https://github.com/mediagis/nominatim-docker) using OpenStreetMap data from GEOFABRIK (https://www.geofabrik.de/) to find locations. Initial testing indicated that the use of full address strings led to false positives, and the computing resource requirement for running the server was high. Thus, only location strings from the second approach that did not lead to country information were used as queries. Because multiple potential matches were returned for each query, the results were sorted based on their location importance values. The above steps led to an additional 72,401 records with country information.

Examining the overlap in country information between approaches revealed that brute force current country and pycountry searches were consistent 97.1% of the time. In addition, both approaches had high consistency with the email-based approach (92.4% and 93.9%). However, brute force subregion and Nominatim-based predictions had the lowest consistencies with the above 3 approaches (39.8% to 47.9%) and each other. Thus, a record's country information was finalized if the information was consistent between any 2 approaches, except between the brute force subregion and Nominatim searches. This led to 330,328 records with country information.

## Topical and country impact metrics

Three metrics were used to access topical and country impacts, namely, SCImago Journal Rank (SJR; [47]), H-index (the number of articles, $h$, in a journal that have received at least $h$ citations), and C-score (citations per document over a 2-year period). These metrics between 1999 and 2020 were obtained from the SCImago Scientific Journal Ranking site (https://www.scimagojr.com/journalrank.php). For each plant science record, the SJR, H-index, or C-score of the journal it was published in was used as the impact score for that record. The overall impact score ($I_{m,t,y}$) of topic $t$ in year $y$ using each impact metric $m$ (**S11 Data**) is defined as

$$I_{m,t,y} = log_2\left(\frac{(\sum_{r=1}^{N_{t,y}} S_{r,t,y})/N_{t,y}}{(\sum_{r=1}^{N_y} S_{r,y})/N_y}\right)$$

where $S_{r,t,y}$ is the impact score of record $r$ in topic $t$ published in year $y$, $N_{t,y}$ is the number of records in $t$ and in $y$, $S_{r,y}$ is the impact score of record $r$ published in year $y$, and $N_y$ is the number of records in $y$. The first average was divided by the second to account for year-to-year differences in overall impact scores. The logarithm was applied to facilitate interpretation of the impact score.

To determine annual country impact, impact scores were determined in the same way as that for annual topical impact, except that values for different countries were calculated instead of topics (**S8 Data**).

## Topical preferences by country

To determine topical preference for a country $C$, a $2 \times 2$ table was established with the number of records in topic $T$ from $C$, the number of records in $T$ but not from $C$, the number of non-$T$ records from $C$, and the number of non-$T$ records not from $C$. A Fisher's exact test was performed for each $T$ and $C$ combination, and the resulting $p$-values were corrected for multiple testing with the Bejamini–Hochberg method (see **S12 Data**). The preference of $T$ in $C$ was defined as the degree of enrichment calculated as log likelihood ratio of values in the $2 \times 2$ table. Topic 5 was excluded because >50% of the countries did not have records for this topic.

The top 10 countries could be classified into a China–India cluster, an Italy–Spain cluster, and remaining countries (yellow rectangles, **Fig 5E**). The clustering of Italy and Spain is partly due to similar research focusing on allergens (topic 0) and mycotoxins (topic 54) and less emphasis on gene family (topic 23) and stress tolerance (topic 28) studies (**Figs 5F and S9**). There are also substantial differences in topical focus between countries. For example, plant science records from China tend to be enriched in hyperspectral imaging and modeling (topic 9), gene family studies (topic 23), stress biology (topic 28), and research on new plant compounds associated with herbal medicine (topic 69), but less emphasis on population genetics and evolution (topic 86, **Fig 5F**). In the US, there is a strong focus on insect pest resistance (topic 75), climate, community, and diversity (topic 83), and population genetics and evolution but less focus on new plant compounds. In summary, in addition to revealing how plant science research has evolved over time, topic modeling provides additional insights into differences in research foci among different countries.

## Supporting information

**S1 Fig. Plant science record classification model performance. (A–C)** Distributions of prediction probabilities (y_prob) of **(A)** positive instances (plant science records), **(B)** negative instances (non-plant science records), and **(C)** positive instances with the Medical Subject Heading "Plants" (ID = D010944). The data are color coded in blue and orange if they are correctly and incorrectly predicted, respectively. The lower subfigures contain log10-transformed $x$ axes for the same distributions as the top subfigure for better visualization of incorrect predictions. **(D)** Prediction probability distribution for candidate plant science records. Prediction probabilities plotted here are available in S13 Data.
(PDF)

**S2 Fig. Relationships between outlier clusters and the 90 topics. (A)** Heatmap demonstrating that some outlier clusters tend to have high prediction scores for multiple topics. Each cell shows the average prediction score of a topic for records in an outlier cluster. **(B)** Size of outlier clusters.
(PDF)

**S3 Fig. Cosine similarities between topics. (A)** Heatmap showing cosine similarities between topic pairs. Top-left: hierarchical clustering of the cosine similarity matrix using the Ward algorithm. The branches are colored to indicate groups of related topics. **(B)** Topic labels and names. The topic ordering was based on hierarchical clustering of topics. Colored rectangles: neighboring topics with >0.5 cosine similarities.
(PDF)

**S4 Fig. Relative topical diversity for 20 journals.** The 20 journals with the most plant science records are shown. The journal names were taken from the journal list in PubMed (https://

www.nlm.nih.gov/bsd/serfile_addedinfo.html).
(PDF)

**S5 Fig. Topical frequency and top terms during different time periods. (A-D)** Different patterns of topical frequency distributions for example topics **(A)** 48, **(B)** 35, **(C)** 27, and **(D)** 42. For each topic, the top graph shows the frequency of topical records in each time bin, which are the same as those in **Fig 3** (green line), and the end date for each bin is indicated. The heatmap below each line plot depicts whether a term is among the top terms in a time bin (yellow) or not (blue). Blue dotted lines delineate different decades (see S5 Data for the original frequencies, S6 Data for the LOWESS fitted frequencies and the top terms for different topics/time bins).
(PDF)

**S6 Fig. Prevalence of records mentioning different taxonomic groups in Viridiplantae. (A, B)** Percentage of records mentioning specific taxa at the **(A)** major lineage and **(B)** family levels. **(C, D)** The prevalence of taxon mentions over time at the **(C)** major lineage and **(E)** family levels. The data used for plotting are available in S9 Data.
(PDF)

**S7 Fig. Changes over time. (A)** Number of genera being mentioned in plant science records during different time bins (the date indicates the end date of that bin, exclusive). **(B)** Numbers of genera (blue) and organisms (salmon) with draft genomes available from National Center of Biotechnology Information in different years. **(C)** Percentage of US National Science Foundation (NSF) grants mentioning the genus *Arabidopsis* over time with peak percentage and year indicated. The data for **(A–C)** are in S9 Data. **(D)** Number of plant science records in the top 17 plant science journals from the USA (red), Great Britain (GBR) (orange), India (IND) (light green), and China (CHN) (dark green) normalized against the total numbers of publications of each country over time in these 17 journals. The data used for plotting can be found in S11 Data.
(PDF)

**S8 Fig. Change in country impact on plant science over time. (A, B)** Difference in 2 impact metrics from 1999 to 2020 for the 10 countries with the highest number of plant science records. **(A)** H-index. **(B)** SCImago Journal Rank (SJR). **(C, D)** Plots show the relationships between the impact metrics (H-index in **(C)**, SJR in **(D)**) averaged from 1999 to 2020 and the slopes of linear fits with years as the predictive variable and impact metric as the response variable for different countries (A3 country codes shown). The countries with >400 records and with <10% missing impact values are included. The data used for plotting can be found in S11 Data.
(PDF)

**S9 Fig. Country topical preference.** Enrichment scores (LLR, log likelihood ratio) of topics for each of the top 10 countries. Red: overrepresentation, blue: underrepresentation. The data for plotting can be found in S12 Data.
(PDF)

**S1 Data. Summary of source journals for plant science records, prediction models, and top Tf-Idf features.** Sheet–Candidate plant sci record j counts: Number of records from each journal in the candidate plant science corpus (before classification). Sheet—Plant sci record j count: Number of records from each journal in the plant science corpus (after classification). Sheet–Model summary: Model type, text used (txt_flag), and model parameters used. Sheet—Model performance: Performance of different model and parameter combinations on the

validation data set. Sheet–Tf-Idf features: The average SHAP values of Tf-Idf (Term frequency-Inverse document frequency) features associated with different terms. Sheet–PubMed number per year: The data for PubMed records in Fig 1A. Sheet–Plant sci record num per yr: The data for the plant science records in Fig 1A.
(XLSX)

**S2 Data. Numbers of records in topics identified from preliminary topic models.** Sheet–Topics generated with a model based on BioBERT embeddings. Sheet–Topics generated with a model based on distilBERT embeddings. Sheet–Topics generated with a model based on Sci-BERT embeddings.
(XLSX)

**S3 Data. Final topic model labels and top terms for topics.** Sheet–Topic label: The topic index and top 10 terms with the highest cTf-Idf values. Sheets– 0 to 89: The top 50 terms and their c-Tf-Idf values for topics 0 to 89.
(XLSX)

**S4 Data. UMAP representations of different topics.** For a topic $T$, records in the UMAP graph are colored red and records not in $T$ are colored gray.
(PDF)

**S5 Data. Temporal relationships between published documents projected onto 2D space.** The 2D embedding generated with UMAP was used to plot document relationships for each year. The plots from 1975 to 2020 were compiled into an animation.
(GIF)

**S6 Data. Timestamps and dates for dynamic topic modeling.** Sheet–bin_timestamp: Columns are: (1) order index; (2) bin_idx–relative positions of bin labels; (3) bin_timestamp–UNIX time in seconds; and (4) bin_date–month/day/year. Sheet–Topic frequency per timestamp: The number of documents in each time bin for each topic. Sheets–LOWESS fit 0.1/0.2/0.3: Topic frequency per timestamp fitted with the fraction parameter of 0.1, 0.2, or 0.3. Sheet —Topic top terms: The top 5 terms for each topic in each time bin.
(XLSX)

**S7 Data. Locally weighted scatterplot smoothing (LOWESS) of topical document frequencies over time.** There are 90 scatter plots, one for each topic, where the $x$ axis is time, and the $y$ axis is the document frequency (blue dots). The LOWESS fit is shown as orange points connected with a green line. The category a topic belongs to and its order in **Fig 3** are labeled on the top left corner. The data used for plotting are in S6 Data.
(PDF)

**S8 Data. The 4 criteria used for sorting topics.** Peak: the time when the LOWESS fit of the frequencies of a topic reaches maximum. 1st_reach_thr: the time when the LOWESS fit first reaches a threshold of 60% maximal frequency (peak value). Trend: upward (1), no change (0), or downward (−1). Stable: whether a topic belongs to the stable category (1) or not (0).
(TXT)

**S9 Data. Change in taxon record numbers and genome assemblies available over time.** Sheet–Genus: Number of records mentioning a genus during different time periods (in Unix timestamp) for the top 100 genera. Sheet–Genus: Number of records mentioning a family during different time periods (in Unix timestamp) for the top 100 families. Sheet–Genus: Number of records mentioning an order during different time periods (in Unix timestamp) for the top 20 orders. Sheet–Species levels: Number of records mentioning 12 selected taxonomic levels

higher than the order level during different time periods (in Unix timestamp). Sheet–Genome assembly: Plant genome assemblies available from NCBI as of October 28, 2022. Sheet–Arabidopsis NSF: Absolute and normalized numbers of US National Science Foundation funded proposals mentioning Arabidopsis in proposal titles and/or abstracts.
(XLSX)

**S10 Data. Taxon topical preference.** Sheet– 5 genera LLR: The log likelihood ratio of each topic in each of the top 5 genera with the highest numbers of plant science records. Sheets– 5 genera: For each genus, the columns are: (1) topic; (2) the Fisher's exact test *p*-value (Pvalue); (3–6) numbers of records in topic *T* and in genus *X* (n_inT_inX), in *T* but not in *X* (n_inT_niX), not in *T* but in *X* (n_niT_inX), and not in *T* and *X* (n_niT_niX) that were used to construct $2 \times 2$ tables for the tests; and (7) the log likelihood ratio generated with the $2 \times 2$ tables. Sheet–corrected *p*-value: The 4 values for generating LLRs were used to conduct Fisher's exact test. The *p*-values obtained for each country were corrected for multiple testing.
(XLSX)

**S11 Data. Impact metrics of countries in different years.** Sheet–country_top25_year_count: number of total publications and publications per year from the top 25 countries with the most plant science records. Sheet—country_top25_year_top17j: number of total publications and publications per year from the top 25 countries with the highest numbers of plant science records in the 17 plant science journals used as positive examples. Sheet–prank: Journal percentile rank scores for countries (3-letter country codes following https://www.iban.com/country-codes) in different years from 1999 to 2020. Sheet–sjr: Scimago Journal rank scores. Sheet–hidx: H-Index scores. Sheet–cite: Citation scores.
(XLSX)

**S12 Data. Topical enrichment for the top 10 countries with the highest numbers of plant science publications.** Sheet—Log likelihood ratio: For each country C and topic T, it is defined as log((a/b)/(c/d)) where a is the number of papers from C in T, b is the number from C but not in T, c is the number not from C but in T, d is the number not from C and not in T. Sheet: corrected *p*-value: The 4 values, a, b, c, and d, were used to conduct Fisher's exact test. The *p*-values obtained for each country were corrected for multiple testing.
(XLSX)

**S13 Data. Text classification prediction probabilities.** This compressed file contains the PubMed ID (PMID) and the prediction probabilities (y_pred) of testing data with both positive and negative examples (pred_prob_testing), plant science candidate records with the MeSH term "Plants" (pred_prob_candidates_with_mesh), and all plant science candidate records (pred_prob_candidates_all). The prediction probability was generated using the Word2-Vec text classification models for distinguishing positive (plant science) and negative (non-plant science) records.
(ZIP)

## Acknowledgments

We thank Maarten Grootendorst for discussions on topic modeling. We also thank Stacey Harmer, Eva Farre, Ning Jiang, and Robert Last for discussion on their respective research fields and input on how to improve this study and Rudiger Simon for the suggestion to examine differences between countries. We also thank Mae Milton, Christina King, Edmond Anderson, Jingyao Tang, Brianna Brown, Kenia Segura Abá, Eleanor Siler, Thilanka Ranaweera, Huan Chen, Rajneesh Singhal, Paulo Izquierdo, Jyothi Kumar, Daniel Shiu, Elliott

Shiu, and Wiggler Catt for their good ideas, personal and professional support, collegiality, fun at parties, as well as the trouble they have caused, which helped us improve as researchers, teachers, mentors, and parents.

## Author Contributions

**Conceptualization:** Shin-Han Shiu, Melissa D. Lehti-Shiu.

**Data curation:** Shin-Han Shiu.

**Formal analysis:** Shin-Han Shiu.

**Funding acquisition:** Shin-Han Shiu.

**Investigation:** Shin-Han Shiu, Melissa D. Lehti-Shiu.

**Methodology:** Shin-Han Shiu.

**Project administration:** Shin-Han Shiu, Melissa D. Lehti-Shiu.

**Resources:** Shin-Han Shiu.

**Software:** Shin-Han Shiu.

**Supervision:** Shin-Han Shiu, Melissa D. Lehti-Shiu.

**Validation:** Shin-Han Shiu.

**Visualization:** Shin-Han Shiu.

**Writing – original draft:** Shin-Han Shiu.

**Writing – review & editing:** Shin-Han Shiu, Melissa D. Lehti-Shiu.

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
