## [Editor Report · Decision Letter 0]

31 Oct 2023

Dear Dr Shiu, 

Thank you for submitting your manuscript entitled "Evolution of research topics and paradigms in a biological field" for consideration as a Research Article by PLOS Biology. Please accept my apologies for the delay incurred while we sought external advice.

Your manuscript has now been evaluated by the PLOS Biology editorial staff, as well as by an academic editor with relevant expertise, and I'm writing to let you know that we would like to send your submission out for external peer review.

IMPORTANT: Please can you change the article type to "Meta-Research Article" when you upload the remaining metadata (see next paragraph)?

Once your full submission is complete, your paper will undergo a series of checks in preparation for peer review. After your manuscript has passed the checks it will be sent out for review. To provide the metadata for your submission, please Login to Editorial Manager (https://www.editorialmanager.com/pbiology) within two working days, i.e. by Nov 02 2023 11:59PM.

Kind regards,

Roli Roberts

Roland Roberts, PhD

Senior Editor

PLOS Biology

rroberts@plos.org

---

## [Decision Letter · Decision Letter 1]

19 Jan 2024

Dear Dr Shiu,

Thank you for your patience while your manuscript "Evolution of research topics and paradigms in a biological field" was peer-reviewed at PLOS Biology. Your manuscript has been evaluated by the PLOS Biology editors, an Academic Editor with relevant expertise, and by two independent reviewers. Please accept our apologies for the additional delay incurred over the holiday period.

As you can imagine, securing reviewers for such an interdisciplinary study was a challenge, but we think that these reviews may prove helpful in guiding your revisions. You'll see that reviewer #1, who has expertise in scientometrics and "big data" methodology, is positive about your approach, which s/he thinks is valuable and original, but says s/he cannot assess the interest to the plant community. Their major concern is that s/he finds their conclusion that there has been a decline in work with Arabidopsis (Fig 4C) very unlikely, which makes them worry about artefacts. S/he also finds the code very inaccessible, and asks you to provide a jupyter notebook to help readers follow it. Their third major concern is that topic grouping should be done via an unbiased objective approach to avoid human bias. S/he has a range of more minor concerns too. Reviewer #2, who is a plant scientist, is rather more sceptical; s/he found the broad findings to be interesting, but thinks that the you may be mistaken in looking for technological explanations for shifts in the field, when conceptual advances (“paradigm shifts”) may be more likely. S/he thinks that while the approach allows one to study the influence of such factors, it may be less good at identifying them a priori.

As you will see in the reviewer reports, which can be found at the end of this email, although the reviewers find the work potentially interesting, they have also raised a substantial number of important concerns. Based on their specific comments and following discussion with the Academic Editor, it is clear that a substantial amount of work would be required to meet the criteria for publication in PLOS Biology. However, given our and the reviewer interest in your study, we would be open to inviting a comprehensive revision of the study that thoroughly addresses all the reviewers' comments. Given the extent of revision that would be needed, we cannot make a decision about publication until we have seen the revised manuscript and your response to the reviewers' comments. Your revised manuscript would need to be seen by the reviewers again, but please note that we would not engage them unless their main concerns have been addressed. 

We appreciate that these requests represent a great deal of extra work, and we are willing to relax our standard revision time to allow you 6 months to revise your study. Please email us (plosbiology@plos.org) if you have any questions or concerns, or envision needing a (short) extension. At this stage, your manuscript remains formally under active consideration at our journal; please notify us by email if you do not intend to submit a revision so that we may withdraw it.

**IMPORTANT - SUBMITTING YOUR REVISION**

*Resubmission Checklist*

*Published Peer Review*

*PLOS Data Policy*

*Blot and Gel Data Policy*

Sincerely,

Roli Roberts

Roland Roberts, PhD

Senior Editor

PLOS Biology

rroberts@plos.org

REVIEWS:

Reviewer's Responses to Questions

PLOS authors have the option to publish the peer review history of their article (what does this mean?). If published, this will include your full peer review and any attached files.

Reviewer #1: No

Reviewer #2: No

Reviewer #1: The article by Shiu et Lehti-Shiu performs a bibliometric analysis of trends of scientific topics in plant research over nearly one century. They further introduce methods and software that could already be applicable to other domains of biomedical research.

Their work extends beyond static work in the biomedical space using BERT models (González-Márquez et al., bioRxiv, 2023) by providing temporal resolution and acting on a scale that, while large, still allows human inspection and interpretability of all topics. Their work extends beyond earlier time-resolved work on trends in the biomedical space (Telis et al., Genetics, 2016) by shifting away from individual keywords toward topics. Besides this unique combination of technology and interpretability, learning about the history of plant science felt enjoyable and valuable, albeit I lack the domain knowledge to confirm the specific insights about plant science.

My primary comment relates to data presented in Figure 4C and its technical implications beyond Figure 4C. Specifically, the authors discuss a decline in Arabidopsis research, which I have personally never observed, and - more importantly - is not seen in PubMed's visualization of the time-resolved popularity of Arabidopsis (graph on left side when querying https://pubmed.ncbi.nlm.nih.gov/?term=arabidopsis&sort=date ). A decline of Arabidopsis research also seems to stand in the opposition to observations of Telis et al., Genetics, 2016, which are based on only one journal (and thus may only provide a partial view).

While Figure 4C shows something unexpected to me, the underlying technical factor may also influence interpretation of other parts of the manuscript, particularly on dynamics.

Though I cannot evaluate from the distance where this difference arises, I suspect false negatives in the creation of the corpus, or a discrepancy between absolute and relative quantifications of occurrences, or a combination of both.

My second comment relates to the re-usability of the work of Shiu et Lehti-Shiu. The individual elements of their code on github look quite nice, and they already structure the code by figures. Given the scale of their code, I however could not find where to start, and in which order to call what script. I would find it very helpful if the authors created a brief jupyter notebook that walks others through their workflow to create data/analysis required for Figs. 1-3, and referred to this notebook in the README.md file in the top of the repository. This notebook could use plants but should convey to people interested in other domains (e.g.: invertebrate biology) how/where to modify code. 

My third comment is that it is presently unclear how topics were grouped in Figure 3, and to which extent this could be the result of human decisions. Preferentially, the grouping would be the result of some unbiased or formalized procedure.

These are my peripheral comments:

Corpus is based on keywords, but usage of keywords has changed, and increased over decades - possibly contributing to growth of the corpus shown in Figure 1A. If not contributing to my primary comment (above), a half sentence and supplemental figure on share of PubMed with keywords could assist interpretation of Figure 1A.

Some of the articles in PubMed do use languages other than English. It is unclear if they would form a separate topic, be dispersed outliers, or - given the usage of BERT - may even appear in topics irrespective of language.

Similarly, it is unclear to which extent the growth reported for China and India is truly exceptional growth of plant research in these countries, or increased publishing in English, or efforts of NCBI to include literature written in languages other than English. While such a distinction could be obtained by analyzing plant-related publications contained in OpenAlex, a few extra words would seem to suffice.

The results section opens with a statement on comparing different models, but the methods section and referred supplemental figure did not allow me to understand, which specific approaches were evaluated and whether differences in performance were significant. Aside from selling them short, inability to follow this initial scientific statement may deter some readers from proceeding to the rest of the publication, which is quite accessible.

Independent of above paragraph, it would be good to see an evaluation of false positive/false negatives in the creation of the corpus. Comparing against a corpus formed by NCBI's MeSH term D010944 (Plants) would seem reasonable as they are an authority and follows good practices. The authors could, for instance, obtain MeSH by running https://github.com/titipata/pubmed_parser on the files that they already downloaded, and obtain the children terms of D010944 via the definition of MeSH terms downloadable on NCBI/MeSH's ftp server.

As Figure 2 and 3 are very information rich, more direct references from text toward figure could assist reading (e.g.: noting that topics 27, 26, 25 are in bottom left corner; or that evolution of topics is seen for topic cluster F shown in blue).

The manuscript would to me seem equally strong if substituting "paradigm shift" by "shift". Similarly, if title was shortened to remove paradigms. Though the authors follow a common-language use of "paradigm shift" this usage does not capture Kuhn's original work, where paradigm shifts are preceded by long periods of stagnation and very rare. This is not in line with the usage of "paradigm shift" as part of Figure 3. Further, Kuhn compared paradigms to declinations in grammar, whereas the present manuscript uses words as a surrogate of paradigms. 

Figure 5D is framed as being "interesting", but would seem to align well with the phenomenon of "regression to the mean".

For Figure 5F it is not clear which of the changes are statistically significant, particularly if controlling for testing of multiple hypotheses.

The focus on topics rather than individual terms promises enhanced robustness of conclusions. However, this is not directly tested. One quick analysis would be to take the top term of each topic in Figure 3, and use it as a surrogate of the topic. Would single terms yield similar observations or very different ones? Either outcome would help others to better grasp the practical consequences of the methodological advances contained in this manuscript.

The discussion section suggests that technological innovation has driven research in plant sciences as key terms frequently are technologies. However, among the 80+ topics, only around 4-6 top terms relate to technologies.

For time resolved binning, find it very clever to bin multiple years prior mid-1990ies to increase signal, but would find it technically safer and more directly human interpretable to use individual years as the resolution of analysis afterwards. This would also dampen technical noise associated with different publication frequencies (Jan 1st often attracts journals/sub-field on annual publication schedule, while other journals/sub-fields are spread out over the year when following monthly schedules).

The projection of publications in a 2D space in Figure 1C is great but made me curious about seeing publication years superimposed. This would help to see how science shifted over the space of the corpus.

Reviewer #2: This paper presents a comprehensive analysis of plant science topics over the last 50 years with a view to identifying key conceptual or technological innovations (paradigm shifts). After arriving at 90 topics, each defined by a combination of about 5 terms, the authors analyse topic frequency over time. They show that some topics have declined, others have had a transitory popularity, and others are on the rise. The most interesting finding is that they identify three major transitions, each involving multiple topics, which they attribute to paradigm shifts at around 1980, 1990 and 2000 (why they happen to be spaced out by 10 years is not discussed). However, when it comes to explaining these shifts, the authors look for largely technical causes, such as adoption of recombinant DNA technologies, better approaches to creating transgenic plants, more efficient mutagenesis etc. Key conceptual changes, w

---

## [Decision Letter · Decision Letter 2]

28 Mar 2024

Dear Dr Shiu,

Thank you for your patience while we considered your revised manuscript "Evolution of research topics in a biological field" for publication as a Meta-Research Article at PLOS Biology. This revised version of your manuscript has been evaluated by the PLOS Biology editors, the Academic Editor and the original reviewers.

Based on the reviews, we are likely to accept this manuscript for publication, provided you satisfactorily address the remaining points raised by the reviewers and the following data and other policy-related requests.

IMPORTANT - please attend to the following:

a) Please change the Title to make it more explicit. We do understand that you wish to present your study as a generalisable approach, with plant science as a case study, but we think that its specific interest to plant scientists would nonetheless be substantial, so we suggest something like: "Using machine learning and language models to assess the evolution of research topics within specific fields: the case of plant science"

b) Please address the remaining concerns raised by the reviewers. Reviewer #1 mentions a few issues where some biases may have crept in. S/he suggests a further small experiment to compare human and machine classifiers and requests some clarifications. Reviewer #2 would like you to discuss an apparent ~10-year periodicity in the data, which s/he wonders might be linked to career cycles.

c) Please address my Data Policy requests below; specifically, we need you to supply the numerical values underlying Figs 1AB, 2B, 3ABCDE, 4ABC, 5ABCDEF, S1ABCD, S5ABCD, S6ABCD, S7ABCD, S8ABCD, S9, either as a supplementary data file or as a permanent DOI’d deposition. I note that you already have a zipped data supplement that contains 12 supp data files (plus other material), but it's unclear how this relates to the Figures. Please clarify the relationship to the Figure panels and/or supply the underlying numerical values.

d) Please cite the location of the data clearly in all relevant main and supplementary Figure legends, e.g. “The data underlying this Figure can be found in S1 Data” or “The data underlying this Figure can be found in https://zenodo.org/records/XXXXXXXX

e) Thank you for making your code available. However, because Github depositions can be readily changed or deleted, please make a permanent DOI’d copy (e.g. in Zenodo) and provide this URL in the manuscript and Data Availability Statement (see below).

We expect to receive your revised manuscript within two weeks. 

*Published Peer Review History*

*Press*

Sincerely,

Roli

Roland Roberts, PhD

Senior Editor

rroberts@plos.org

PLOS Biology

DATA POLICY:

Regardless of the method selected, please ensure that you provide the individual numerical values that underlie the summary data displayed in the following figure panels as they are essential for readers to assess your analysis and to reproduce it: Figs 1AB, 2B, 3ABCDE, 4ABC, 5ABCDEF, S1ABCD, S5ABCD, S6ABCD, S7ABCD, S8ABCD, S9. NOTE: the numerical data provided should include all replicates AND the way in which the plotted mean and errors were derived (it should not present only the mean/average values).

CODE POLICY

Per journal policy, if you have generated any custom code during the curse of this investigation, please make it available without restrictions upon publication. Please ensure that the code is sufficiently well documented and reusable, and that your Data Statement in the Editorial Manager submission system accurately describes where your code can be found.

DATA NOT SHOWN?

REVIEWERS' COMMENTS:

Reviewer #1:

The authors have largely addressed my comments. This is an improvement of a strong manuscript, and I especially wish to thank the authors for the added _examples in their code repository. 

My prior main critique was the reported drop in Arabidopsis thaliana research. The language of the revised manuscript now is more precise to convey that this is a relative statement, relative to all other plant research. Further, I can confirm in an own reanalysis of PubMed, that there is a drop relative to other plant publications, when latter are assessed in the reference journals provided by the authors. 

As the manuscript singles out Arabidopsis thaliana, a possible reading is that there is something special going on with Arabidopsis thaliana. However, it is unclear to me if these observations are specific to Arabidopsis thaliana, or a (relative) decrease of model system research, or a (relative) decrease of specific plant species being mentioned in abstracts. I suggest a word of caution or one control analysis.

A preceding peripheral point of mine is the estimation of the false-positive and false-negative rates. The authors now include an analysis. Based on the methods section it appears that these rates are estimated by testing consistency with the authors' respective training / testing sets, and/or cross-validation on held-out samples. While this strategy suggests consistency, it however does not tell if the authors' assumptions on plant and not-plant are correct. Further, these numbers could be biased when applied to the corpus of reference journals of which most publications should be about plants. 

I suggest that the authors additionally randomly 50 papers labelled as "Plant" by their classifier, 50 random papers labelled as "not-plant", mix both groups of papers, only keep title and abstract, and have human curator blinded toward "Plant" and "not-plant" (and journal names of these papers etc.) label as "Plant" and "Not-Plant". Then compare human and computational labelling in a confusion matrix, and estimate overall number of false positives through Bayes' theorem, to account for plant science only being a minority of publications on PubMed.

Another peripheral point is the usage of "keywords" to define the corpus. As stated by the authors in their reply they used the title or abstract to define concepts, not PubMed keywords. Along this point, I suggest further editing for clarity line 74 which speaks about keywords: "To this end, we first collected over 30 million PubMed records and narrowed down candidate plant science records with plant-related terms and taxon names as keywords"

Reviewer #2:

The authors have addressed my concerns and the paper is now a more measured and better qualified description of their findings. I wonder if the authors could comment on the 10 year cycle which they also find curious. Could it be that this relates not only to the scientific advances but also to the fashion-driven aspect of science. That is, for research to sound current and novel (important both for grants and publications), it needs to latch on to new buzz words. As these new words become prevalent, yet new buzz words are needed. So the ten year cycle could reflect, at least in part, the time it takes for buzz-word to peak and saturate. Given that a PhD takes around 4-5 years, a grant is typically 3-5 years, and a scientific career lasts around 30-40 years, a 10 year cycle would correspond to two PhDs, 2-3 grants, or one third or quarter of a scientific career. According to this hypothesis, similar cycle durations would be predicted for other scientific disciplines, which could be tested through further studies.

---

## [Editor Report · Decision Letter 3]

4 Apr 2024

Dear Dr Shiu,

Thank you for the submission of your revised Meta-Research Article "Assessing the evolution of research topics in a biological field using plant science as an example" for publication in PLOS Biology. On behalf of my colleagues and the Academic Editor, Ulrich Dirnagl, I'm pleased to say that we can in principle accept your manuscript for publication, provided you address any remaining formatting and reporting issues. These will be detailed in an email you should receive within 2-3 business days from our colleagues in the journal operations team; no action is required from you until then. Please note that we will not be able to formally accept your manuscript and schedule it for publication until you have completed any requested changes.

Sincerely, 

Roli Roberts

Senior Editor

PLOS Biology

rroberts@plos.org